# Quantifying the shape of cells, from Minkowski tensors to p-atic orders

Lea Happel[1†], Griseldis Oberschelp[1†], Valeriia Grudtsyna[2], Harish P Jain[3], Rastko Sknepnek[4,5], Amin Doostmohammadi[2], Axel Voigt[1,6,7]*

[1]Institute of Scientific Computing, Dresden, Germany; [2]Niels Bohr Institute, University of Copenhagen, Kobenhavn, Denmark; [3]Njord Centre, Physics Department, University of Oslo, Oslo, Norway; [4]School of Life Sciences, University of Dundee, Dundee, United Kingdom; [5]School of Science and Engineering, University of Dundee, Dundee, United Kingdom; [6]Center of Systems Biology Dresden, Dresden, Germany; [7]Cluster of Excellence, Physics of Life, TU Dresden, Dresden, Germany

## eLife Assessment

This **important** work describes a set of parameters that give a robust description of shape features of cells in tissues. The evidence for the usefulness of these parameters is **solid**. The work should be of interest for anybody analyzing epithelial dynamics, but more details about the analysis of experimental images are necessary and some streamlining of the text would increase the accessibility of the material for non-specialists.

**\*For correspondence:**
axel.voigt@tu-dresden.de

[†]These authors contributed equally to this work

**Competing interest:** The authors declare that no competing interests exist.

**Abstract** *P*-atic liquid crystal theories offer new perspectives on how cells self-organize and respond to mechanical cues. Understanding and quantifying the underlying orientational orders is, therefore, essential for unraveling the physical mechanisms that govern tissue dynamics. Due to the deformability of cells this requires quantifying their shape. We introduce rigorous mathematical tools and a reliable framework for such shape analysis. Applying this to segmented cells in MDCK monolayers and computational approaches for active vertex models and multiphase field models allows to demonstrate independence of shape measures and the presence of various *p*-atic orders at the same time. This challenges previous findings and opens new pathways for understanding the role of orientational symmetries and *p*-atic liquid crystal theories in tissue mechanics and development.

## Introduction

The importance of orientational order in biological systems is becoming increasingly clear, as it plays a critical role in processes such as tissue morphogenesis, collective cell motion, and cellular extrusion. Orientational order results from the shapes of cells and their alignments with neighboring cells. Disruptions in this order, known as topological defects, are often linked to key biological events. For instance, defects—points or lines where the order breaks down—can drive cell extrusion (*Saw et al., 2017*; *Monfared et al., 2023*) or trigger morphological changes in tissues (*Maroudas-Sacks et al., 2021*; *Ravichandran et al., 2025*). Orientational order is linked to liquid crystal theories (*de Gennes, 1993*) and should here be interpreted in a broad sense. Recent evidence extends beyond nematic order, characterized by symmetry under $180° = 2\pi/2$ rotation, to higher-order symmetries such as tetratic order ($90° = 2\pi/4$), hexatic order ($60° = 2\pi/6$), and even general *p*-atic orders ($2\pi/p$, *p* being an integer). In biological contexts, nematic order ($p = 2$) has been widely studied in epithelial tissues (*Duclos et al., 2017*; *Saw et al., 2017*; *Kawaguchi et al., 2017*), linking defects to cellular behaviors and tissue organization. More complex orders, such as tetratic order ($p = 4$) (*Cislo et al.,*

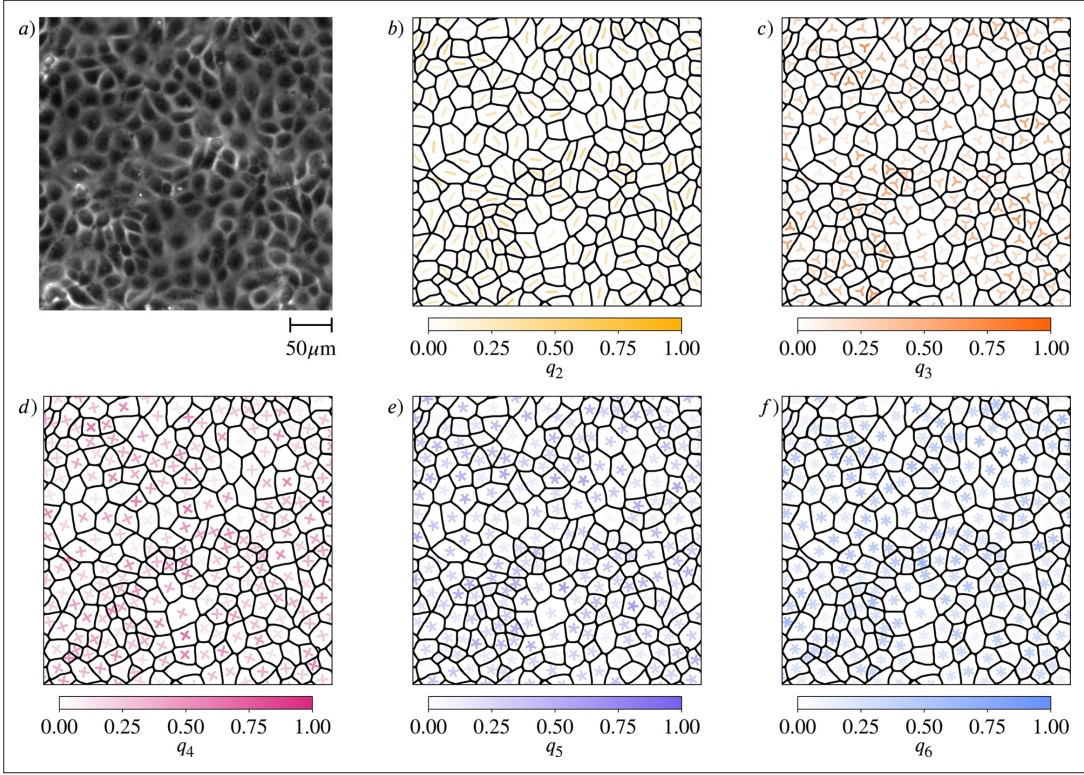

**Figure 1.** Shape classification of cells in wild-type Madin-Darby canine kidney (MDCK) cell monolayer. (**a**) Raw experimental data. (**b-f**) Minkowski tensor, visualized using $\vartheta_p$ and $q_p$, *Equation 7* (see Methods) for $p = 2, 3, 4, 5, 6$, respectively. The brightness and the rotation of the $p$-atic director indicates the magnitude and the orientation, respectively. The visualization uses rotationally symmetric direction fields known as $p$-RoSy fields in computer graphics (*Vaxman et al., 2016*). See Appendix 1-Experimental setup for details on the experimental data.

*2023*) and hexatic oder ($p = 6$) (*Li and Ciamarra, 2018*; *Durand and Heu, 2019*; *Pasupalak et al., 2020*; *Armengol-Collado et al., 2023*; *Eckert et al., 2023*), have also been observed in experimental systems, offering new perspectives on how cells self-organize and respond to mechanical cues. Understanding and quantifying orientational order and the corresponding liquid crystal theory are, therefore, essential for unraveling the physical mechanisms that govern biological dynamics.

In physics, $p$-atic liquid crystals illustrate how particle shape and symmetry influence phase behavior, with seminal works dating back to Onsager's theories (*Onsager, 1949*). Most prominently, hexatic order ($p = 6$) has been postulated and found by experiments and simulations as an intermediate state between crystalline solid and isotropic liquid in *Halperin and Nelson, 1978*; *Nelson and Halperin, 1979*; *Murray and van Winkle, 1987*; *Bladon and Frenkel, 1995*; *Zahn et al., 1999*; *Gasser et al., 2010*; *Bernard and Krauth, 2011*. Other examples are colloidal systems for triadic platelets (*Bowick et al., 2017*) or cubes (*Wojciechowski and Frenkel, 2004*), which lead to $p$-atic order with $p = 3$ and $p = 4$, respectively. Even pentatic ($p = 5$) and heptatic ($p = 7$) liquid crystals have been engineered (*Wang and Mason, 2018*; *Yu and Mason, 2024*). The corresponding liquid crystal theories for $p$-atic order have only recently been defined (*Giomi et al., 2022b*; *Krommydas et al., 2023*) and can also be used in biological contexts. However, while in colloidal systems particle shapes remain fixed, in biological tissues, cells are dynamic: their shapes are irregular, variable, and influenced by internal and external forces. These unique properties make quantifying $p$-atic order in tissues significantly more challenging, as quantification of the cell shapes is required. We will demonstrate that existing methods, such as bond-orientational order (*Nelson and Halperin, 1979*) or polygonal shape analysis (*Armengol-Collado et al., 2023*), might fail to capture the nuances of irregular cell shapes, which has severe consequences on the definition of $p$-atic order.

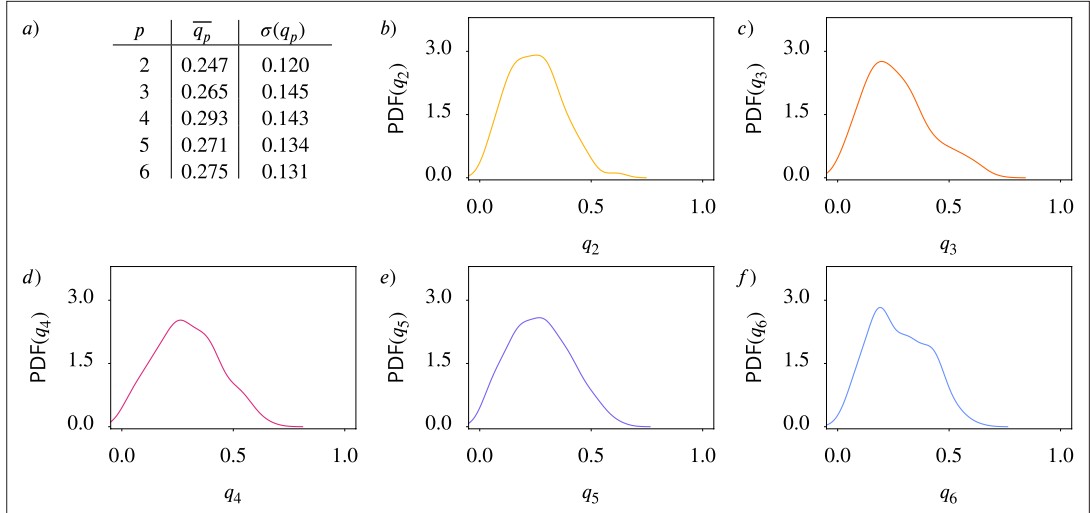

**Figure 2.** Statistical data for cell shapes identified in *Figure 1* (see Methods). (**a**) Mean $\overline{q_p}$ and standard deviation $\sigma(q_p)$ of $q_p$. (**b- f**) Probability distribution function (PDF) of $q_p$ for $p = 2, 3, 4, 5, 6$, respectively. Kde-plots are used to show the probability distribution.

To address this, we adapt Minkowski tensors (*Mecke, 2000*; *Schröder-Turk et al., 2013*)—rigorous mathematical tools for shape analysis—to quantify *p*-atic orders in cell monolayers. Minkowski tensors provide robust and sensitive measures of shape anisotropy and orientation, accommodating both smooth and polygonal shapes while remaining resilient to small perturbations. By applying these tools, we re-examine previous conclusions about *p*-atic orders in epithelial tissues and demonstrate that certain widely accepted results require reconsideration.

Our study leverages a combination of experimental and computational approaches. Experimentally, we analyze confluent monolayers of MDCK (Madin-Darby Canine Kidney) cells. *Figure 1* provide a snapshot of a considered monolayer of 235 wild-type MDCK cells, together with their shape classification by Minkowski tensors. The corresponding statistical data and probability distributions of these quantities are shown in *Figure 2*. These data indicate the presence of all *p*-atic orders at once with similar probability distributions, mean values, and standard deviations. In addition to these experiments, we also analyze data of MDCK cells reported in *Armengol-Collado et al., 2023*. Computationally, we employ two complementary models for cell monolayers: the active vertex model and the multiphase field model. These approaches allow us to systematically vary parameters such as cell activity and mechanical properties, providing a comprehensive view of how *p*-atic orders emerges in different contexts.

By combining these robust mathematical tools, experimental insights, and computational models, we establish a reliable framework for quantifying *p*-atic orders in biological tissues. This framework not only challenges previous findings, e.g. a proposed hexatic-nematic crossover on larger length scales (*Armengol-Collado et al., 2023*), but also opens new pathways for understanding the role of orientational symmetries in tissue mechanics and development, which require to consider multiple orientational symmetries.

## Methods
### Minkowski tensors for shape classification
Essential properties of the geometry of a two-dimensional object are summarized by scalar-valued size measures, so called Minkowski functionals. They are defined by

$$W_0(\mathcal{C}) = \int_{\mathcal{C}} d\mathcal{C}, \quad W_1(\mathcal{C}) = \int_{\partial\mathcal{C}} d\partial\mathcal{C}, \text{ and } \quad W_2(\mathcal{C}) = \int_{\partial\mathcal{C}} \mathcal{H} \, d\partial\mathcal{C}, \tag{1}$$

where $\mathcal{C}$ is the smooth two-dimensional object, with contour $\partial\mathcal{C}$ and $\mathcal{H}$ denotes the curvature of the contour $\partial\mathcal{C}$ (see *Figure 3* for a schematic description), as reviewed in *Mecke, 2000*. They describe

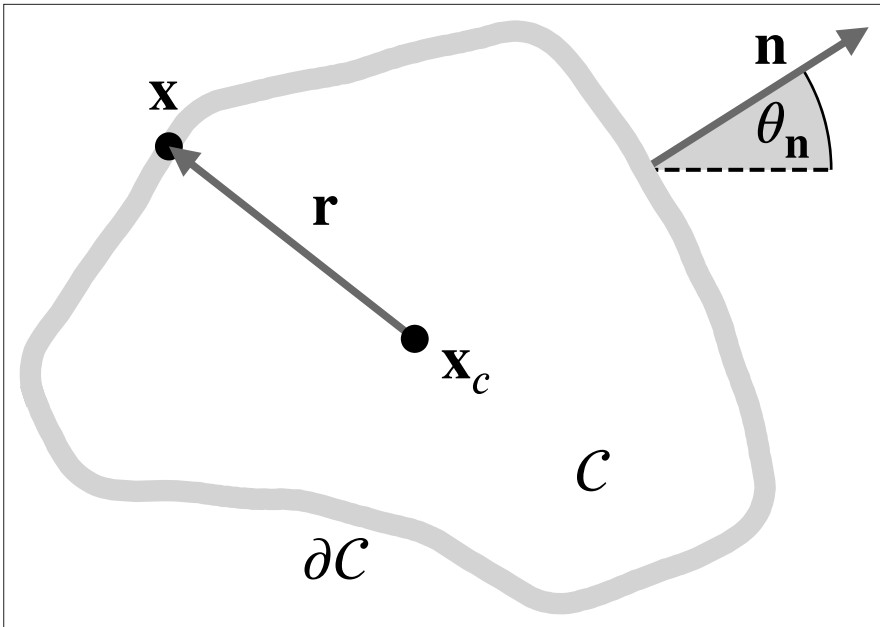

**Figure 3.** Schematic description of a two-dimensional object $\mathcal{C}$ with contour $\partial\mathcal{C}$. We denote the center of mass with $\mathbf{x}_c$ and vectors from $\mathbf{x}_c$ to points $\mathbf{x}$ on $\partial\mathcal{C}$ with $\mathbf{r}$. The outward-pointing normals are denoted by $\mathbf{n}$, the corresponding angle with the $x$-axis by $\theta_{\mathbf{n}}$.

the area, the perimeter, and a curvature-weighted integral of the contour, respectively. Minkowski functionals are also known as intrinsic volumes. For convex objects, they have been shown to be continuous and invariant to translations and rotations. Due to these properties and Hadwiger's characterization theorem (*Hadwiger, 1975*), they are natural size descriptors that provide essential and complete information about invariant geometric features. However, these properties also set limits to their use as shape descriptors. Due to rotation invariance, they are unable to capture the orientation of the shape. To describe more complex shape information, the scalar-valued Minkowski functionals are extended to a set of tensor-valued descriptors, known as Minkowski tensors. These objects have been investigated both in mathematical (*Alesker, 1999*; *Hug et al., 2008a*; *Hug et al., 2008b*; *McMullen, 1997*) and in physical literature (*Schröder-Turk et al., 2010b*; *Kapfer, 2012*). Using tensor products of the position vectors $\mathbf{r}$ and the normal vectors $\mathbf{n}$ of the contour $\partial\mathcal{C}$, defined as $\mathbf{r}^a \odot \mathbf{n}^b = \mathbf{r} \odot \ldots \odot \mathbf{r} \odot \mathbf{n} \odot \ldots \odot \mathbf{n}$, the first considered $a$ times and the last $b$ times with $\odot$ denoting the symmetrized tensor product, the Minkowski tensors are defined as

$$\mathbf{W}_0^{a,0}(\mathcal{C}) = \int_{\mathcal{C}} \mathbf{r}^a \, \mathrm{d}\mathcal{C}, \quad \mathbf{W}_1^{a,b}(\mathcal{C}) = \int_{\partial\mathcal{C}} \mathbf{r}^a \odot \mathbf{n}^b \, \mathrm{d}\partial\mathcal{C}, \quad \text{and} \quad \mathbf{W}_2^{a,b}(\mathcal{C}) = \int_{\partial\mathcal{C}} \mathbf{r}^a \odot \mathbf{n}^b \mathcal{H} \, \mathrm{d}\partial\mathcal{C}. \tag{2}$$

For $\mathbf{W}_j^{0,0}(\mathcal{C}) = W_j(\mathcal{C})$ for $j = 0, 1, 2$, so the Minkowski tensors are extensions of the Minkowski functionals. In analogy to Minkowski functionals, also Minkowski tensors have been shown to be continuous for convex objects. Also, analogies to Hadwiger's characterization theorem exist (*Alesker, 1999*). However, for our purpose, the essential property is the continuity. It makes Minkowski functionals and Minkowski tensors a robust measure as small shape changes lead to small changes in the shape descriptor. Due to this property, Minkowski tensors have been successfully used as shape descriptors in different fields, e.g., in materials science as a robust measure of anisotropy in porous (*Schröder-Turk et al., 2010a*) and granular material (*Schröder-Turk et al., 2013*), in astrophysics to describe morphology of galaxies (*Beisbart, 2002*), and in biology to distinguish shapes of different types of neuronal cell networks (*Beisbart et al., 2006*) and to determine the direction of elongation in multiphase field models for epithelial tissue (*Mueller et al., 2019*; *Wenzel and Voigt, 2021*; *Happel and Voigt, 2024*). However, the mentioned examples exclusively use lower-rank Minkowski tensors $a + b \leq 2$. Higher rank tensors have not been considered in such applications, but the theory also guarantees robust description of $p$-atic orientation for $p = 3, 4, 5, 6, \ldots$. These results hold for smooth

contours but can also be extended to polyconvex shapes. Furthermore, known counterexamples for non-convex shapes have very little to no relevance in applications (*Kapfer et al., 2012*).

Even if all Minkowski tensors carry important geometric information, one type is particularly interesting and will be considered in the following:

$$W_1^{0,p}(\mathcal{C}) = \int_{\partial\mathcal{C}} \underbrace{\mathbf{n} \odot \mathbf{n} \odot \ldots \odot \mathbf{n}}_{p-times} \, d\partial\mathcal{C}. \tag{3}$$

In this case, the symmetrized tensor product agrees with the classical tensor product. If applied to polygonal shapes the Minkowski tensors $W_1^{0,p}(\mathcal{C})$ are related to the Minkowski problem for convex polytops (i.e. generalizations of three-dimensional polyhedra to an arbitrary number of dimensions), which states that convex polytops are uniquely described by the outer normals of the edges and the length of the corresponding edge (*Minkowski, 1897*). Several generalizations of this result exist (*Klain, 2004*; *Schneider, 2013*), which makes the normal vectors a preferable quantity to describe shapes. However, Minkowski tensors contain redundant information, which asks for an irreducible representation. This can be achieved by decomposing the tensor with respect to the rotation group $SO(2)$ (*Kapfer, 2012*; *Mickel et al., 2013*; *Klatt et al., 2022*). Following this approach, one can write

$$W_1^{0,p}(\mathcal{C}) = \int_{S^1} \mathbf{u}^p \Psi_{\mathcal{C}}(\mathbf{u}) \, d\mathbf{u} \quad \text{with} \quad \Psi_{\mathcal{C}}(\mathbf{u}) = \int_{\partial\mathcal{C}} \delta(\mathbf{n}(\mathbf{x}) - \mathbf{u}) \, d\partial\mathcal{C}, \tag{4}$$

with $\Psi_{\mathcal{C}}(\mathbf{u})$ being proportional to the probability density of the normal vectors. Identifying $\mathbf{u} \in S^1$ by the angle $\theta$ between $\mathbf{u}$ and the $x$-axis allows to write

$$\Psi_{\mathcal{C}}(\theta) = \sum_{p=-\infty}^{\infty} \psi_p(\mathcal{C}) e^{-ip\theta} \quad \text{with} \quad \psi_p(\mathcal{C}) = \frac{1}{2\pi} \int_0^{2\pi} \Psi_{\mathcal{C}}(\theta) e^{ip\theta} \, d\theta. \tag{5}$$

The Fourier coefficients $\psi_p(\mathcal{C})$ are the irreducible representations of $W_1^{0,p}(\mathcal{C})$ and can be written as

$$\psi_p(\mathcal{C}) = \frac{1}{2\pi} \int_{\partial\mathcal{C}} e^{ip\theta_n} \, d\partial\mathcal{C}, \tag{6}$$

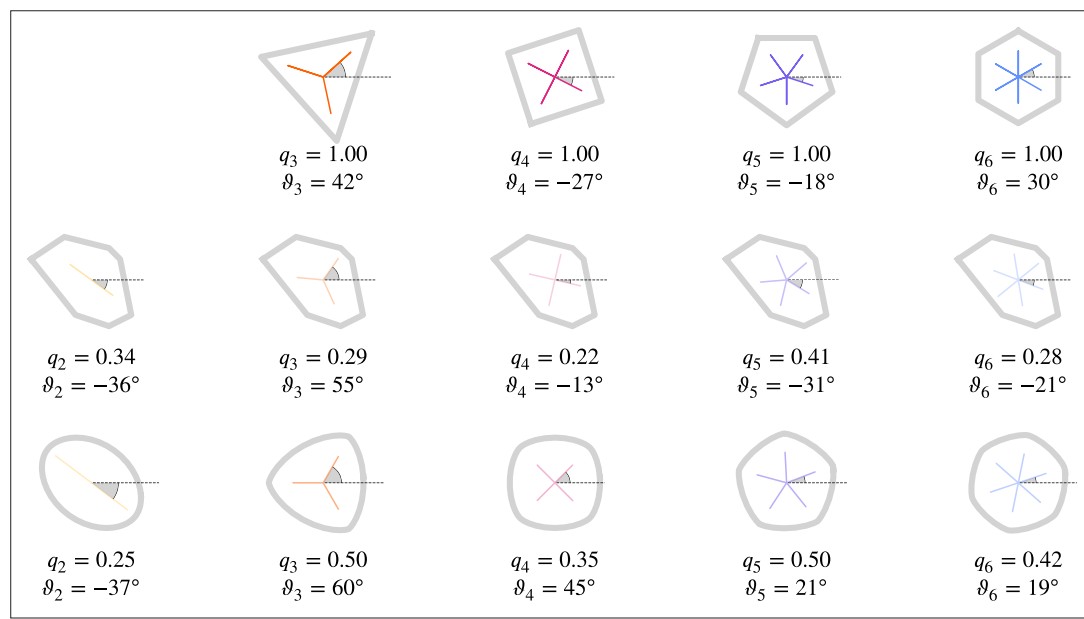

**Figure 4.** Regular and irregular shapes, adapted from *Armengol-Collado et al., 2023* and *Schaller et al., 2024*, by *Equation 7*. For regular shapes, the corresponding magnitude of $q_p$ is always $1.0$ and the detected angle is the minimal angle of the $p$-atic orientation with respect to the $x$-axis. Note that no shape with $q_2 = 1.0$ is shown, as this would be a line. The visualization is according to *Figure 1*.

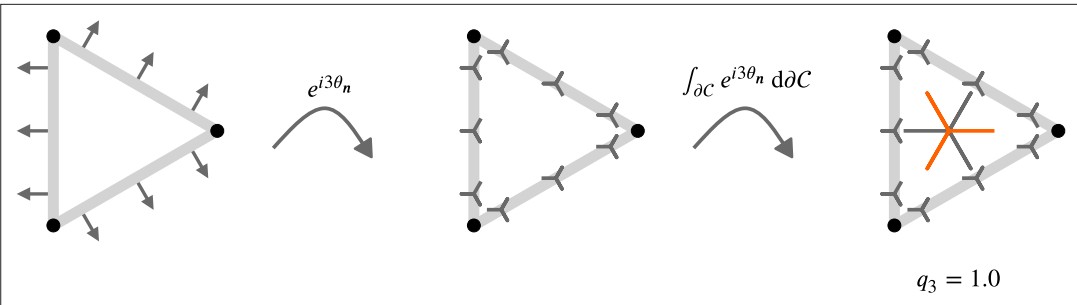

**Figure 5.** Illustrative description of the definition of $q_3$ for an equilateral triangle. Considering rotational symmetries under a rotation $120° = 2\pi/3$ means that vectors with an angle of $120°$ or $240°$ are treated as equal. Applied to the normals $\mathbf{n}$ (left), this means that under this rotational symmetry, the normals on the three different edges are equal. Mathematically, this is expressed through $e^{3i\theta_{\mathbf{n}}}$ resulting in the triatic director shown instead of the normals $\mathbf{n}$ (middle). One leg of the triatic director always points in the direction as the normal. While only shown for three points on each edge, we obtain an orientation with the respective symmetry on every point of the contour $\partial\mathcal{C}$. Considering the line integral along the contour provides the dominating triadic director, shown in the center of mass (right). To get a value between $0.0$ and $1.0$ for $q_3$, we normalize this integral with the length of the contour, which corresponds to $q_0$. As all triatic directors point in the same direction, we obtain $q_3 = 1.0$ in this specific example. To be consistent with other approaches, we rotate the resulting triadic director by $60° = \pi/3$ leading to the orange triadic director, which is the quantity used for visualization.

where $\theta_{\mathbf{n}}$ is the orientation of the outward pointing normal $\mathbf{n}$. The phase of the complex number $\psi_p(\mathcal{C})$ contains information about the preferred orientation and the absolute value $|\psi_p(\mathcal{C})|$ is a scalar index. We thus define

$$\vartheta_p(\mathcal{C}) = \frac{1}{p}\arctan 2\left(\Im\psi_p(\mathcal{C}), \Re\psi_p(\mathcal{C})\right) + \frac{\pi}{p} \quad \text{and} \quad q_p(\mathcal{C}) = \frac{|\psi_p(\mathcal{C})|}{\psi_0(\mathcal{C})}, \tag{7}$$

with $\Im\psi_p$ and $\Re\psi_p$ the imaginary and real part of $\psi_p$, respectively, and $\psi_0(\mathcal{C}) = \frac{W_1(\mathcal{C})}{2\pi}$. We have $q_p(\mathcal{C}) \in [0, 1]$ quantifying the strength of $p$-atic order. *Figure 4* illustrates the concept for polygonal and smooth shapes and *Figure 5* illustrates the calculation of $q_p$ and $\vartheta_p$ at the example of an equilateral triangle. Note that the constant factor of $1/(2\pi)$ in *Equation 6* does not influence the result of *Equation 7* and is, therefore, disregarded in *Figure 5*. Alternative derivations of *Equation 7* consider higher order trace-less tensors (*Virga, 2015*; *Giomi et al., 2022b*; *Armengol-Collado et al., 2023*) generated by the normal $\mathbf{n}$. In this case $\vartheta_p(\mathcal{C})$ (orange triatic director in *Figure 5* right) corresponds to the eigenvector to the negative eigenvalue and $\vartheta_p(\mathcal{C}) - \pi/p$ (gray triatic director in *Figure 5* right) to the eigenvector to the positive eigenvalue. For $p = 2$, this tensor-based approach corresponds to the structure tensor (*Mueller et al., 2019*).

## Alternative shape measures

As pointed out in the introduction, $p$-atic orders have already been identified in biological tissue and model systems. However, the way to quantify rotational symmetry in these works differs from the Minkowski tensors. In *Armengol-Collado et al., 2023*, the cell contour is approximated by a polygon with $V$ vertices having coordinates $\mathbf{x}_i$. The considered polygonal shape analysis is based on the shape function

$$\gamma_p(C) = \frac{\sum_{i=1}^{V} |\mathbf{r}_i|^p e^{ip\theta_i}}{\sum_{i=1}^{V} |\mathbf{r}_i|^p}, \tag{8}$$

with $\mathbf{r}_i = \mathbf{x}_i - \mathbf{x}_c$ being the vector from the centroid $\mathbf{x}_c$ and $\theta_i$ being the orientation of the $i$-th vertex of the polygon with respect to its center of mass. As the centroid is computed by $\mathbf{x}_c = \frac{1}{V}\sum_{i=1}^{V}\mathbf{x}_i$ *Armengol-Collado et al., 2023* the only input data are the coordinates of the vertices of the $V$-sided polygon. *Equation 8* captures the degree of regularity of the polygon by its amplitude $0 \le |\gamma_p| \le 1$ and its orientation $\vartheta_p^{\gamma}$ which follows as

| $q_3 = 0.980$ | $q_3 = 0.999$ | $q_3 = 1.000$ | $|\gamma_3| = 0.900$ | $|\gamma_3| = 0.927$ | $|\gamma_3| = 1.000$ |
| $q_4 = 0.033$ | $q_4 = 0.002$ | $q_4 = 0.000$ | $|\gamma_4| = 0.556$ | $|\gamma_4| = 0.495$ | $|\gamma_4| = 0.000$ |

**Figure 6.** Defining *p*-atic order for deformable objects requires robust shape descriptors. Shown is the strength of *p*-atic order for a polygon converging to an equilateral triangle. (**a**) using $q_p$ and (**b**) using $\gamma_p$. The considered vectors used in the computations, normals **n** of the contour for the Minkowski tensors and $\mathbf{r}_i$ for $\gamma_p$, are shown. Note that the removal of the forth vertex highly influences the value of $\gamma_p$. How $\mathbf{x}_c$ is calculated - as the mean of the vertex coordinates or as the center of mass of the polygon - can also slightly alter the results. We used the described approach following *Armengol-Collado et al., 2023*.

$$\vartheta_p^\gamma(C) = \frac{1}{p} \arctan \frac{\Im \gamma_p(C)}{\Re \gamma_p(C)}, \tag{9}$$

with $\Im \gamma_p$ and $\Re \gamma_p$ the imaginary and real part of $\gamma_p$, respectively. Instead of the normals **n**, the descriptor is based on the position vector **r**. One might be tempted to think that *Equation 9* is related to the irreducible representation of $W_1^{p,0}$. This is, however, not the case, as the weighting in *Equation 8* is done with respect to the magnitude of **r**, while in $W_1^{p,0}$ the weighting is done with respect to the contour integral. Furthermore, in contrast to the Minkowski tensors, *Equation 8* takes into account only the vectors at the vertices and not the vectors along the full cell contour. While this seems to be a technical detail, it has severe consequences. The theoretical basis, which guarantees continuity, no longer holds. This leads to unstable behaviour, as illustrated in *Figure 6*. While the shape of the polygons is almost identical, there is a jump in $\gamma_3$ and $\gamma_4$ as we go from four to three vertices. The weighting according to the magnitude cannot cure this, as the magnitude of the shrinking vector is far from zero shortly before this vertex vanishes. In the context of cellular systems, such deformations are a regular occurrence rather than an artificially constructed test case. We will demonstrate the impact while discussing the results and recommend only using robust descriptors such as the Minkowski tensors or their irreducible representations.

We further note that bond order parameters $\Psi_p^{bond}$, which consider the connection between cells, have also been used as shape descriptors (*Li and Ciamarra, 2018*; *Durand and Heu, 2019*; *Pasupalak et al., 2020*). Introduced in *Nelson and Halperin, 1979*, these parameters are

$$\Psi_p^{bond} = \frac{1}{B} \sum_{b=1}^{B} e^{ip\theta_b}, \tag{10}$$

with $B$ denoting the number of bonds and $\theta_b$ the orientation of the bond $b$. In the context of monolayer tissues, $B$ is understood as the number of neighbors, and $\theta_b$ as the orientation of the connection of the center of mass of the current cell with the center of mass of the neighbor $b$ (*Loewe et al., 2020*; *Monfared et al., 2023*). So, cells with the same contour but different neighbor relations lead to different bond order parameters. This characteristic should already disqualify these measures as shape descriptors. However, as discussed in detail in *Mickel et al., 2013*, they are not robust even for the task they are designed for. We, therefore, do not discuss them further. The same argumentation holds for other measures which are based on connectivity, as, e.g., considered in *Graner et al., 2008*; *Merkel et al., 2017*.

## Coarse-grained quantities

We define coarse-grained quantities, following closely the strategy used in *Armengol-Collado et al., 2023*. We, therefore, regard the coarse-grained strength of *p*-atic order $Q_p = Q_p(\mathbf{x})$, which is the average of all shape functions $\frac{\psi_p}{\psi_0}$ (or equivalently $q_p e^{ip\vartheta_p}$) of cells whose center of mass $\mathbf{x}_c$ lies within a circle with radius $R$ and center **x**. In a formula, this is:

$$Q_P(\mathbf{x}) = \frac{\sum_{j=1}^{N} q_P(\mathcal{C}_j) e^{ip\vartheta_p(\mathcal{C}_j)} \Theta(R - |\mathbf{x} - \mathbf{x}_c(\mathcal{C}_j)|)}{\sum_{j=1}^{N} \Theta(R - |\mathbf{x} - \mathbf{x}_c(\mathcal{C}_j)|)} \tag{11}$$

where $N$ is the number of cells. $\mathcal{C}_i$ denotes the $i$-th cell and $\Theta$ denotes the Heaviside step function with $\Theta(x) = 1$ for $x > 0$ and $\Theta(x) = 0$ otherwise. As *Armengol-Collado et al., 2023* the position $\mathbf{x}$ is sampled over a square grid with a spacing close to the mean cell radius $R_{cell}$. We calculate this as $R_{cell} = \sqrt{\frac{A_{\mathcal{C}}}{\pi}}$ with $A_{\mathcal{C}}$ the cell area. *Equation 11* provides the basis for validation of continuous $p$-atic liquid crystal theories on the tissue scale, e.g., (*Giomi et al., 2022b*; *Giomi et al., 2022a*). For comparison, we also consider the coarse-grained shape function $\Gamma_p = \Gamma_p(\mathbf{x})$, which is the average of all shape functions $\gamma_p$, which has been considered in *Armengol-Collado et al., 2023* and reads

$$\Gamma_p(\mathbf{x}) = \frac{\sum_{j=1}^{N} \gamma_p(\mathcal{C}_j) \Theta(R - |\mathbf{x} - \mathbf{x}_c(\mathcal{C}_j)|)}{\sum_{j=1}^{N} \Theta(R - |\mathbf{x} - \mathbf{x}_c(\mathcal{C}_j)|)}. \tag{12}$$

Following *Armengol-Collado et al., 2023* $\mathbf{x}_c$ is calculated as $\mathbf{x}_c = \frac{1}{V} \sum_{i=1}^{V} \mathbf{x}_i$ for $\Gamma_p$.

While $Q_p$ and $\Gamma_p$ allow to analyze clustering on the tissue scale, their averages $\overline{Q_p}$ and $\overline{\Gamma_p}$ are tightly related to the statistical properties of the probability distributions of $q_p$ and $|\gamma_p|$. We consider these properties only for comparison and follow the method used in *Armengol-Collado et al., 2023*: At first, we calculate for every time instance/frame the spatial means, $Q_p^t$ and $\Gamma_p^t$, by averaging over all grid points. Then we calculate $\overline{Q_p}$ and $\overline{\Gamma_p}$ by averaging in time, so averaging over all $Q_p^t$ and $\Gamma_p^t$. As in *Armengol-Collado et al., 2023*, the s.e.m. and the standard deviation refer to the averaging in time. $\overline{Q_p}$ and $\overline{\Gamma_p}$ will be used for the discussion of a proposed hexatic-nematic crossover at larger length scales (*Armengol-Collado et al., 2023*).

## Results

Quantifying orientational order in biological tissues can be realized by Minkowski tensors. The orientation $\vartheta_p$ and the strength of $p$-atic order $q_p$ in *Equation 7* can be computed for each cell. Minkowski tensors provide reliable quantities describing how cell shapes align with specific rotational symmetries. As already indicated in *Figure 2*, situations might occur in which rotational symmetries cannot be associated with one specific $p$, but various symmetries seem to be present at the same time. One might be tempted to compare the probability distribution functions (PDFs) of $q_p$ or the mean values $\overline{q_p}$ for different $p$ in order to identify a dominating $p$-atic order. This is particularly important for interpreting nematic ($p = 2$) and hexatic ($p = 6$) orders, which describe distinct symmetries but have been found to coexist in biological systems. However, a direct comparison of these quantities only makes sense if they are comparable to each other. As already mentioned in *Figure 4*, this is not the case, as, e.g., $q_2 = 1.0$ cannot be realized for a cell with a given area. Another question one might ask is if these values are independent of each other. To answer this question, we first address statistically if $q_2$ and $q_6$, evaluated for each cell, are independent. Second, we explore how $q_p$ depends on key parameters determining tissue mechanics. In a third step, we coarse-grain these quantities using the measures $Q_2$ and $Q_6$ in *Equation 11*. With these quantities, we address a proposed hexatic-nematic crossover in epithelial tissue, where hexatic order dominates at small scales and nematic order prevails at larger scales (*Eckert et al., 2023*; *Armengol-Collado et al., 2023*; *Armengol-Collado et al., 2024*). For these three tasks, we analyze simulation data from two computational models (see Appendix 1 including additional references *Hakim and Silberzan, 2017*; *Alert and Trepat, 2020*; *Moure and Gomez, 2021*; *Koride et al., 2018*; *Das et al., 2021*; *Killeen et al., 2022*), the active vertex model (Appendix 1 -Active vertex model, including additional references *Honda, 1983*; *Honda, 2022*; *Farhadifar et al., 2007*; *Bi et al., 2015*; *Tong et al., 2023*) and the multiphase field model (Appendix 1 - Multiphase field model, including additional references *Vey and Voigt, 2007*; *Witkowski et al., 2015*; *Praetorius and Voigt, 2018*; *Salvalaglio et al., 2021*). Although these models differ conceptually, both have been validated in studies of cell monolayer mechanics, including solid-liquid transitions, neighbor

exchange dynamics, and stress profiles (*Fletcher et al., 2014*; *Alt et al., 2017*; *Li et al., 2019*; *Balasubramaniam et al., 2021*; *Wenzel and Voigt, 2021*; *Sknepnek et al., 2023*; *Melo et al., 2023*). Key model parameters are the deformability and the activity strength. They are crucial for capturing the coarse-grained properties of confluent tissues (*Jain et al., 2023*; *Jain et al., 2024*). In the active vertex model, deformability is controlled by the target shape index $p_0$, reflecting the balance between cell-cell adhesion and cortical tension. The multiphase field model, by contrast, encodes deformability through the capillary number $Ca$, which directly incorporates cortical tension. We vary activity strength ($v_0$), the shape index ($p_0$), and the capillary number ($Ca$), ensuring all parameter combinations remain in the fluid regime. Fluidity was confirmed using by mean square displacement (MSD) (*Loewe et al., 2020*), neighbor number variance (*Wenzel and Voigt, 2021*), and the self-intermediate scattering function (*Bi et al., 2016*).

These studies demonstrate independence of $q_2$ and $q_6$, a general trend of increasing $\overline{q_2}$ and decreasing $\overline{q_6}$ for higher activity and deformability, and a consistent decrease of $\overline{Q_2}$ and $\overline{Q_6}$ for increasing coarse-graining radius $R$. However, as $q_2$ and $q_6$ are not directly comparable and $q_2$ and $q_6$ (and, therefore, also $Q_2$ and $Q_6$) are independent, the concept of a hexatic-nematic transition - typically requiring a single order parameter - may not be applicable in this context. Even if the proposed hexatic-nematic crossover (*Armengol-Collado et al., 2023*) is not a formal phase transition, one might expect it to be quantifiable. Yet, its characterization appears to depend strongly on the maximum attainable value of $q_2$, which in turn is influenced by several parameters. To further explore this, we reanalyze the experimental data for MDCK cells from *Armengol-Collado et al., 2023* using Minkowski tensors and compute $q_2$ and $q_6$ as defined in *Equation 7* (see Appendix 1-Experimental setup for details on the analysis of this experimental data with additional references *Stringer et al., 2021*; *van der Walt et al., 2014*). Our analysis also indicates independence of $q_2$ and $q_6$ supporting the same interpretation as above. The statistical properties and probability distributions of these data remain consistent when full cellular boundaries from microscopy images are used. However, an increase in hexatic order ($p = 6$) is observed when cell shapes are approximated by polygons. This suggests that the dominant hexatic order reported at the cellular scale in *Armengol-Collado et al., 2023* may stem from the geometric simplification of cell boundaries. We also compute $Q_2$ and $Q_6$ (*Equation 11*) and again observed a consistent decrease with increasing coarse-graining radius $R$, without evidence of a measurable hexatic-nematic crossover. To better understand the differences with the findings in *Armengol-Collado et al., 2023*, we further analyze both simulation and experimental data using the alternative shape measures $\gamma_p$ (*Equation 8*) and $\Gamma_p$ (*Equation 12*) considered in *Armengol-Collado et al., 2023*. These measures reproduce the reported results.

## Independence of $q_2$ and $q_6$

We examine the distribution of ($q_2$, $q_6$) values across deformability-activity parameter pairs in both computational models (*Figure 7*). Both models show consistent trends: In near-solid regimes (*Figure 7* lower left), $q_2$ and $q_6$ values cluster tightly due to restricted shape fluctuations. However, even in this regime, small $q_2$ values can correspond to either small or large $q_6$ values, and vice versa. In more fluid-like regimes, with higher activity and higher deformability (*Figure 7* upper right), $q_2$ and $q_6$ values become highly scattered. Each $q_2$ value spans a broad range of $q_6$ values, and vice versa, indicating their independence. In order to quantify this, we compute the distance correlation (*Székely et al., 2007*), which is a statistical measure quantifying linear and non-linear dependency in given data. Thereby, a value of 0.0 corresponds to independence, whereas a value of 1.0 corresponds to a strong dependence between the datasets. As can be seen in *Figure 7—figure supplement 1* the obtained distance correlation for $q_2$ and $q_6$ is quite low, underscoring that these quantities are independent. Furthermore, the corresponding p-values, as shown in *Figure 7—figure supplement 2* are mostly larger than 0.1, indicating that the weak correlation found in *Figure 7—figure supplement 1* is not significant. This leads to the conclusion that $q_2$ and $q_6$ measure distinct aspects of cell shape anisotropy. As a consequence, both orders, nematic and hexatic, need to be considered independently. There cannot be a single parameter which describes a crossover between both.

A full investigation of potential dependencies between $q_p$ for arbitrary combinations of $p$'s resulting, e.g., from symmetry arguments is beyond the scope of this paper.

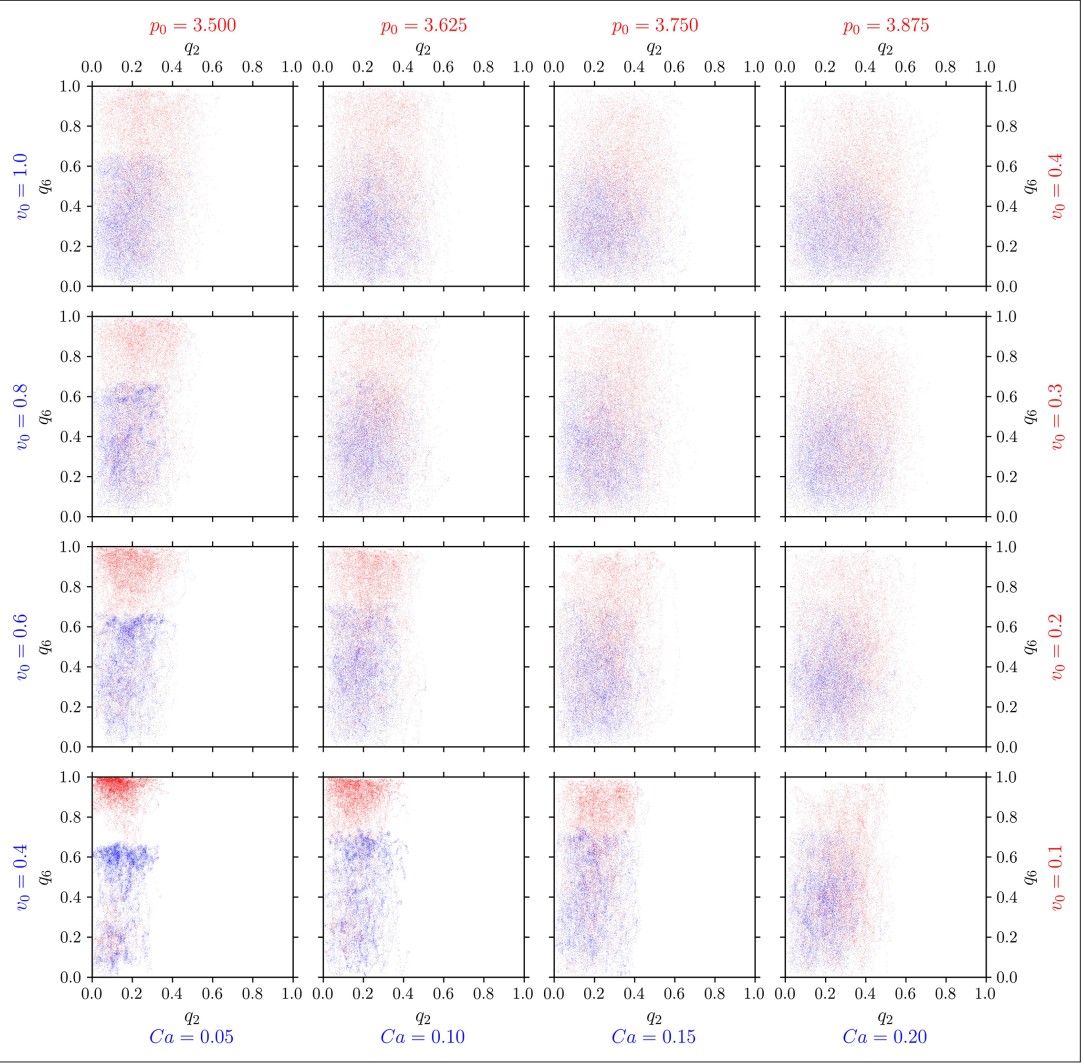

**Figure 7.** Nematic ($p = 2$) and hexatic ($p = 6$) orders are independent of each other. $q_6$ (y-axis) versus $q_2$ (x-axis) for all cells in the multiphase field model (blue) and active vertex model (red). For each cell and each timestep, we plot one point $(q_2, q_6)$. Each panel corresponds to specific model parameters; $Ca$ and $v_0$ for multiphase field model, and $p_0$ and $v_0$ for the active vertex model, representing deformability and activity, respectively.

The online version of this article includes the following figure supplement(s) for figure 7:

**Figure supplement 1.** Distance correlation $dCor(q_2, q_6)$ between $q_2$ and $q_6$ for all cells in the multiphase field model (MPF - purple box) and active vertex model (AV - green box).

**Figure supplement 2.** P-values of the distance correlation $P_{dCor(q_2, q_6)}$ between $q_2$ and $q_6$ for all cells in the multiphase field model (MPF - purple box) and active vertex model (AV - green box).

## Dependence of $q_p$ on activity and deformability

We now explore how tissue properties such as activity and deformability influence cell shape and orientational order. We again focus on nematic ($p = 2$) and hexatic ($p = 6$) orders, as shown in *Figure 8—figure supplements 1 and 2*. Additional results for $p = 3, 4, 5$ are provided in Appendix 2-Results for $q_3$, $q_4$, and $q_5$, precisely in *Appendix 2—figure 1* - *Appendix 2—figure 3* (fixed activity) and *Appendix 2—figure 4*, *Appendix 2—figure 5*, *Appendix 2—figure 6* (fixed deformability). In all plots, all cells and all time steps are considered. In *Figure 8—figure supplement 1*, we vary deformability ($p_0$ in the active vertex model and $Ca$ in the multiphase field model) while keeping the activity $v_0$ constant. Results are presented for the active vertex model (left column) and the multiphase field model (right column). Activity increases from bottom to top rows. Both models show qualitatively similar trends in the probability distribution functions (PDFs) of $q_2$ and $q_6$. For $p = 6$,

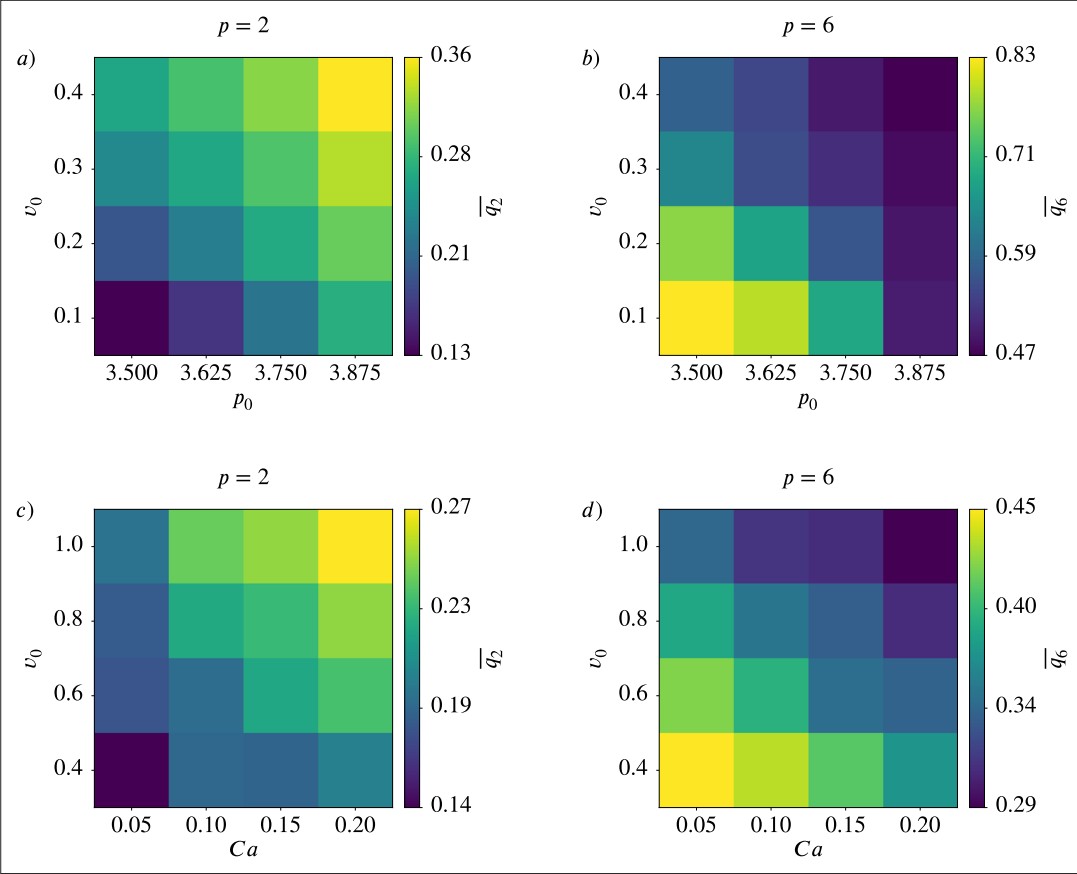

**Figure 8.** Nematic ($p = 2$) and hexatic ($p = 6$) order depend on activity and deformability of the cells. Mean value $\overline{q_p}$ for $p = 2$ (left) and $p = 6$ (right) as function of deformability $p_0$ or $Ca$ and activity $v_0$ for active vertex model (**a** and **b**) and multiphase field model (**c** and **d**).

The online version of this article includes the following figure supplement(s) for figure 8:

**Figure supplement 1.** Nematic ($p = 2$) and hexatic ($p = 6$) order depend on deformability of the cells.

**Figure supplement 2.** Nematic ($p = 2$) and hexatic ($p = 6$) order depend on activity of the cells.

increasing deformability shifts the PDF of $q_6$ to the left, indicating lower mean values. In contrast, for $p = 2$, higher deformability leads to higher $q_2$ values, reflected in a rightward shift of the PDF. This trend is confirmed by the mean values, shown as a function of deformability ($p_0$ and $Ca$, respectively) in the inlets. Additionally, the PDFs broaden with increasing deformability, and this effect is more pronounced at lower activity levels. A notable difference between the models is the range of $q_p$ values. In the multiphase field model, $q_p$ rarely exceeds 0.8 due to the smoother, more rounded cell shapes, whereas the active vertex model often produces higher $q_p$ values. In **Figure 8—figure supplement 2**, deformability is held constant while activity $v_0$ is varied. Results are again shown for the active vertex model (left column) and the multiphase field model (right column), with increasing activity indicated by brighter colors. Deformability increases from bottom to top rows. The trends mirror those observed in **Figure 8—figure supplement 1**. Increasing activity reduces the mean value of $q_6$ while increasing $q_2$, see inlets. The effects of activity are more pronounced at lower deformabilities; at higher deformabilities, differences between parameter regimes diminish. The overall behavior of $q_2$ and $q_6$ is summarized in **Figure 8**, which shows the mean values $\overline{q_2}$ and $\overline{q_6}$ as functions of activity and deformability. For both models:

- $\overline{q_2}$ increases with higher activity or deformability
- $\overline{q_6}$ decreases with higher activity or deformability.

This behavior is consistent with a trend towards nematic order in more dynamic regimes and towards hexatic order in more constrained, less dynamic regimes. The first is further confirmed by

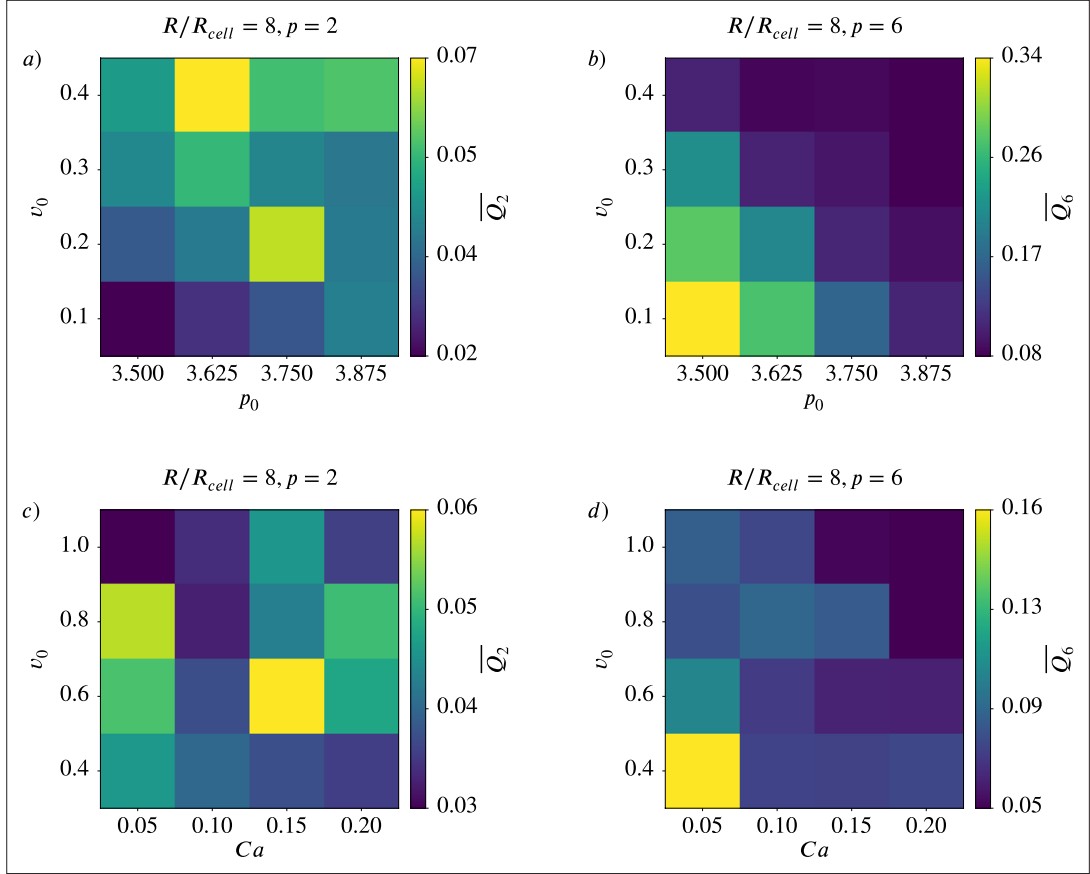

**Figure 9.** Coarse-gained nematic ($p = 2$) and hexatic ($p = 6$) order for $R/R_{cell} = 8$ depend on activity and deformability of the cells. Mean value $\overline{Q_p}$ for $p = 2$ (left) and $p = 6$ (right) as function of deformability $p_0$ or $Ca$ and activity $v_0$ for active vertex model ($a$ and $b$) and multiphase field model ($c$ and $d$).

The online version of this article includes the following figure supplement(s) for figure 9:

**Figure supplement 1.** $\overline{Q_6}$ versus $\overline{Q_2}$ for different coarse-graining radii in the active vertex model.

**Figure supplement 2.** $\overline{Q_6}$ versus $\overline{Q_2}$ for different coarse-graining radii in the multiphase field model.

recent findings in analysing T1 transitions and their effect on cell shapes (*Jain et al., 2025*). These studies suggest that cells transiently elongate when they are undergoing T1 transitions. As the number of T1 transitions increases with activity or deformability *Jain et al., 2024*, this elongation contributes to the observed behavior. The second is consistent with the emergence of hexagonal arrangements in solid-like states.

Corresponding results for $p = 3, 4, 5$ are shown in *Appendix 2—figure 7*. While $q_3$, $q_4$, $q_5$ also increase with increasing activity or deformability, the dependency is not as pronounced as for $q_2$.

## Coarse-grained quantities $Q_2$ and $Q_6$ and potential hexatic-nematic crossover

For every parameter configuration in the active vertex model (*Figure 9—figure supplement 1*) and in the multiphase field model (*Figure 9—figure supplement 2*) we compute the coarse-grained quantities $Q_2$ and $Q_6$ for various coarse-graining radii $R$. One might ask the question if the observed trends for $\overline{q_2}$ and $\overline{q_6}$ for higher activity and deformability in *Figure 8* are also present on larger scales. We, therefore, investigate the behavior of $\overline{Q_2}$ and $\overline{Q_6}$ upon varying activity and deformability, see *Figure 9*. This is exemplified for $R/R_{cell} = 8.0$. The trends seen for $\overline{q_2}$ and $\overline{q_6}$ - so increasing ($\overline{q_2}$) or decreasing ($\overline{q_6}$) with higher activity or deformability - are lost for $\overline{Q_2}$ and less pronounced for $\overline{Q_6}$.

In *Armengol-Collado et al., 2023* a similar approach was used to identify a hexatic-nematic crossover (see *Armengol-Collado et al., 2023*, *Figure 3e*). This cross-over is considered at the

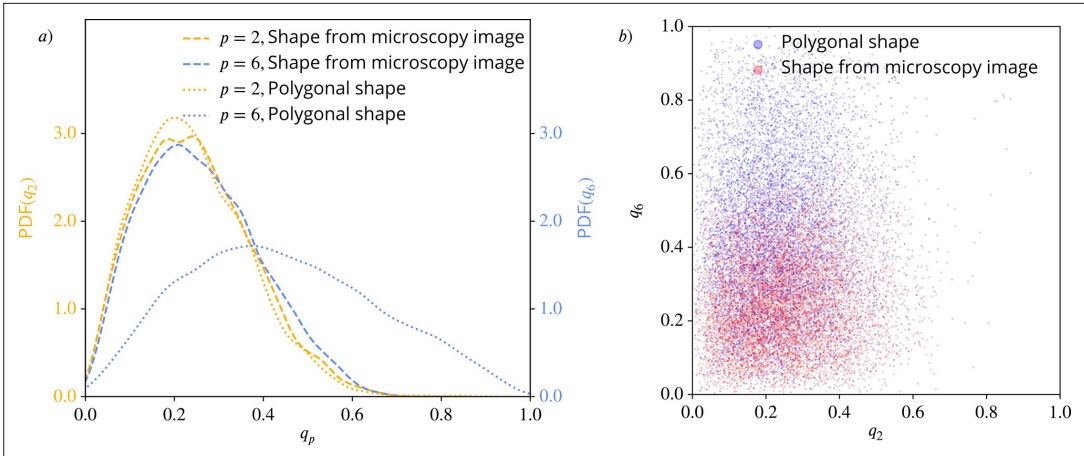

**Figure 10.** Nematic ($p = 2$) and hexatic ($p = 6$) order for the cells in the experiments from *Armengol-Collado et al., 2023*. (**a**) Probability distribution functions (PDFs) using kde-plots, for $q_2$ (yellow) and $q_6$ (blue), once using the polygonal approximation of the cell shape and once using the detailed cell outline obtained from the microscopy pictures. (**b**) $q_6$ (y-axis) versus $q_2$ (x-axis) for all cells from the experimental data in *Armengol-Collado et al., 2023*, once using the polygonal approximation of the cell shape (blue) and once using the detailed cell outline obtained from the microscopy pictures (red). For each cell and each timestep, we plot one point ($q_2, q_6$).

The online version of this article includes the following figure supplement(s) for figure 10:

**Figure supplement 1.** Experimental image, segmented cell outline and polygonal shape.

**Figure supplement 2.** Distance correlation $dCor(q_2, q_6)$ between $q_2$ and $q_6$.

**Figure supplement 3.** P-values of the distance correlation $P_{dCor(q_2, q_6)}$ between $q_2$ and $q_6$.

coarse-graining radius at which the two curves for $\overline{Q_2}$ and $\overline{Q_6}$ as a function of $R$ cross. While already conceptually questioned above, we consider these investigations to compare with *Armengol-Collado et al., 2023*. However, regardless of the model, there is no consistent trend indicative of a potential crossover. These results question the proposed hexatic-nematic crossover reported in *Armengol-Collado et al., 2023*. To further explore this issue, we next test the existence of such a crossover directly on the data considered in *Armengol-Collado et al., 2023*.

## Analyzing experimental data for MDCK cells

The experimental data for confluent monolayers of MDCK GII cells used in *Armengol-Collado et al., 2023* are provided in two different formats, as microscopy images and as polygon data with the calculated vertex points per cell. We consider both formats and all 68 provided configurations. The polygonal data are directly used to compute $q_2$ and $q_6$. The experimental microscopy images are segmented and the extracted cell boundaries are used to compute $q_2$ and $q_6$.

We compute the PDFs of $q_2$ and $q_6$, *Figure 10a* for both data sets. While they are similar, if the full cellular contour from the microscopy images is considered the PDFs strongly differ if the polygonal shapes are used. The dominating hexatic ($p = 6$) order is, therefore, just a consequence of the approximation of the cell boundaries by polygons. This different behavior, which shows larger values for $q_6$ for the polygonal shapes has the same origin as the difference between the regular and rounded shapes in *Figure 4*. Further differences result from the different accessible parameter range. While $q_6 = 1.0$ is possible for a perfect hexagon, $q_2 = 1.0$ cannot be realized, as this would correspond to a line.

We also test the values for $q_2$ and $q_6$ for independence, see *Figure 10b* and in the corresponding statistical measures, *Figure 10—figure supplements 2 and 3*. The indicated independence in the scatter plots *Figure 10b* is confirmed by the distance correlation and the p-values, as in *Figure 7*, *Figure 7—figure supplement 1* and *Figure 7—figure supplement 2*. This holds for the polygonal shapes as well as for the more detailed shapes from the microscopy images.

As a consequence, the same arguments as discussed above also hold for the experimental data and thus caution against interpretation of $q_2$ and $q_6$ or their coarse-grained quantities $Q_2$ and $Q_6$ as

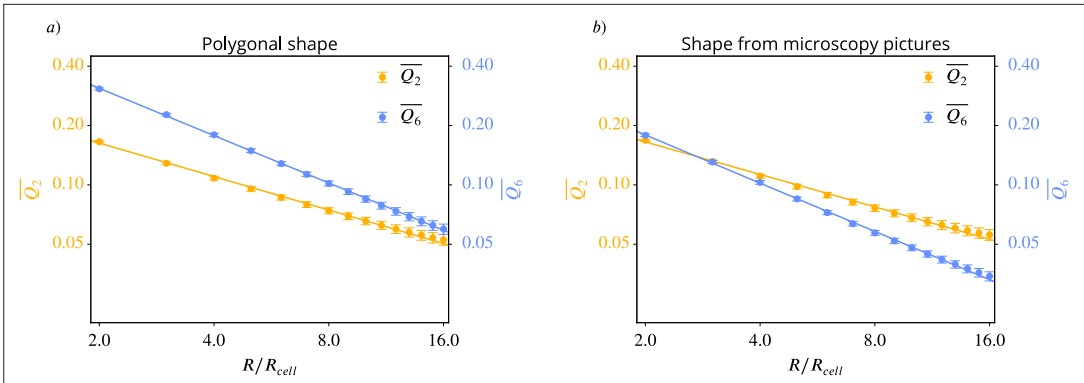

**Figure 11.** $\overline{Q_6}$ versus $\overline{Q_2}$ for different coarse-graining radii for the experimental data from *Armengol-Collado et al., 2023*. On the left side (**a**) we use the polygonal approximation of the cell shape, on the right side (**b**) we use the detailed cell outline obtained from the microscopy pictures. $Q_p$ was calculated according to *Equation 11*, the averaging of this and the choice of $R_{cell}$ follow the description in Coarse-grained quantities. The maximal coarse-graining radius corresponds to half the domain width. A logarithmic scaling was used for both axes. Error bars are obtained as s.e.m.

interdependent order parameters. However, in order to compare with *Armengol-Collado et al., 2023* we next compute the averaged coarse-grained quantities $\overline{Q_2}$ and $\overline{Q_6}$ for various coarse-graining radii $R$. In *Figure 11* these curves are shown for the polygonal shapes (a) and the microscopy images (b). As in *Armengol-Collado et al., 2023*, we carried out the coarse-graining until the coarse-graining radius corresponds to half of the domain width. In both plots, the curves for $\overline{Q_2}$ are almost identical, reflecting the similar PDFs in *Figure 10a*. For $\overline{Q_6}$ the slope is similar but the curves are shifted. A potential crossover, therefore, also depends on the approximation of the cells. In any case, for the considered data, no consistent hexatic-nematic crossover can be observed.

In order to resolve the discrepancy of these results with *Armengol-Collado et al., 2023* we next examine the analysis using the alternative shape measures $\gamma_p$ in *Equation 8*, which have been considered in *Armengol-Collado et al., 2023* but are shown to be not stable.

## Sensitivity of the results on the considered shape descriptor

We now demonstrate that the alternative shape descriptors $\gamma_p$ in *Equation 8*, which have been used in *Armengol-Collado et al., 2023*, can lead to qualitatively different and thus misleading results. The corresponding figures to *Figures 1 and 2* and *Figure 4* are shown in Appendix 2 -Results using polygonal shape analysis, precisely in *Appendix 2—figure 9*, *Appendix 2—figure 10* and *Appendix 2—figure 8*, respectively. For the experimental data in *Figures 1 and 2* we use the Voronoi interface method (*Saye and Sethian, 2011*; *Saye and Sethian, 2012*) to calculate the vertices of a polygon

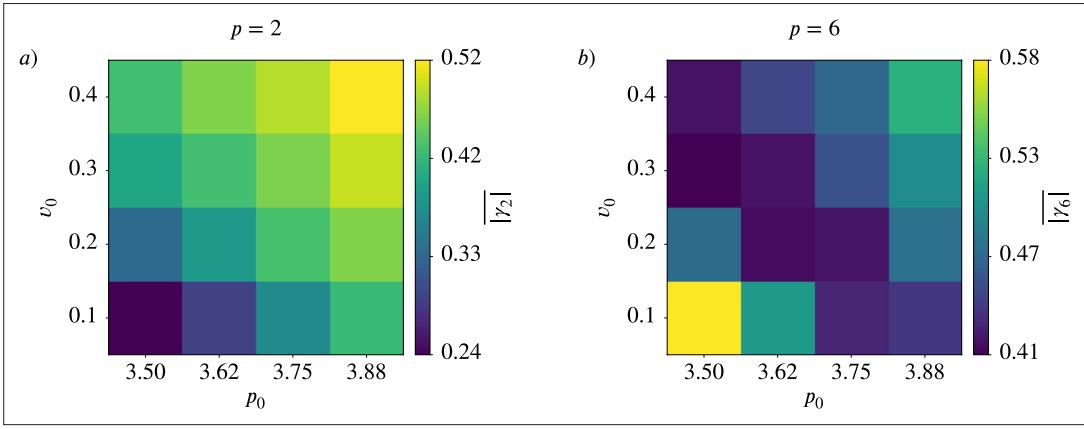

**Figure 12.** Mean value $\overline{|\gamma_p|}$ as function of deformability $p_0$ and activity $v_0$ for active vertex model. (**a**) nematic order ($p = 2$), (**b**) hexatic order ($p = 6$).

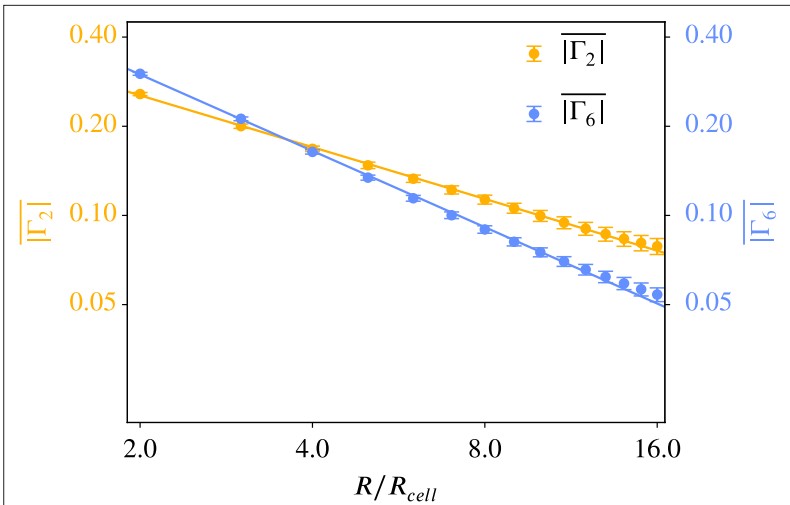

**Figure 13.** $\overline{|\Gamma_6|}$ versus $\overline{|\Gamma_2|}$ for different coarse-graining radii for the experimental data from **Armengol-Collado et al., 2023**. We use only the polygonal approximation of the cell shape as $\gamma_p$ can only work with polygons. $Q_p$ was calculated according to **Equation 12**, the averaging of this and the choice of $R_{cell}$ follow the description in Coarse-grained quantities. The maximal coarse-graining radius corresponds to half the domain width. A logarithmic scaling was used for both axes. Error bars are obtained as s.e.m.

approximating the cell shape. The comparison of these figures already indicates differences between the two methods. Such differences can be seen in the approximated cell shapes, the PDFs and more quantitatively also by comparing $\overline{|\gamma_p|}$ in **Appendix 2—figure 10a** with $\overline{q_p}$ in **Figure 2a**, which, e.g., for $p = 6$ almost double.

For a more detailed comparison of $|\gamma_p|$ and $q_p$ we investigate the data from the active vertex model and the polygonal approximation of the cells in **Armengol-Collado et al., 2023**. We restrict ourselves to this data, as for multiphase field data, the usage of $\gamma_p$ first requires the approximation of a cell by a polygon and we have already seen that this approximation strongly influences the results.

We focus on **Figure 8** and the corresponding results in **Figure 12**. Instead of the monotonic trend for activity and deformability, which was found for $\overline{q_6}$, $\overline{|\gamma_6|}$ exhibits non-monotonic trends, peaking at intermediate values of activity or deformability. For completeness, we also provide the corresponding figures to **Figures 7 and 9** in **Appendix 2—figures 11 and 12**, respectively.

For the polygonal data of the MDCK cells considered in **Armengol-Collado et al., 2023**, we compare the coarse-grained quantities $\overline{Q_2}$ and $\overline{Q_6}$, already considered in **Figure 11a**, with $\overline{|\Gamma_2|}$ and $\overline{|\Gamma_6|}$ computed from $\gamma_2$ and $\gamma_6$ using **Equation 12**, see **Figure 13**. For completeness, we also provide the corresponding figure to **Figure 10** in **Appendix 2—figure 13**. Besides minor differences regarding the calculation of $R_{cell}$ **Figure 13** corresponds to **Armengol-Collado et al., 2023, Figure 3e**. Comparing **Figure 11a** and **Figure 13** one might come to the conclusion that there is no hexatic-nematic crossover using $\overline{Q_2}$ and $\overline{Q_6}$, but there is a hexatic-nematic crossover using $\overline{|\Gamma_2|}$ and $\overline{|\Gamma_6|}$. As the only difference between these two evaluations is the considered shape characterization, this analysis adds another argument that the proposed crossover is not a robust physical feature of the system.

These results also demonstrate that not only the approximation of the cell boundaries by polygonal shapes heavily influences the characterization of $p$-atic order but also the considered method to classify the shape might lead to qualitative different results. This confirms the argumentation in Methods that Minkowski tensors should be preferred because of their stability properties.

## Discussion

In this study, we introduced Minkowski tensors as a robust and versatile tool for quantifying $p$-atic order in multicellular systems, particularly in scenarios involving rounded or irregular cell shapes. By applying this framework to extensive datasets from two distinct computational models—the active vertex model and the multiphase field model—we identified universal trends: increasing activity and deformability of the cells enhance nematic order ($p = 2$) while diminishing hexatic order ($p = 6$). The

consistency of these findings across two models, despite their inherent differences, underscores the generality of our results.

While various shape characterization methods, such as the bond order parameter (*Loewe et al., 2020*; *Monfared et al., 2023*) and the shape function $\gamma_p$ (*Armengol-Collado et al., 2023*), have been explored in the literature, we demonstrated that the choice of shape descriptor significantly impacts the conclusions drawn. Such divergences, together with limited mathematical foundations, highlight the limitations of these alternative shape measures in capturing consistent patterns and emphasize the need for stable, reliable shape measures like the Minkowski tensors. As the stability of Minkowski tensors - in contrast to the bond order parameter or the shape function $\gamma_p$ - can be mathematically justified Minkowski tensors should be the preferred shape descriptor. This finding is not merely a technical nuance, it leads to qualitative differences. Analyzing experimental data for MDCK cells, e.g., has demonstrated that a strong hexatic order on the cellular scale has no physical origin but is a consequence of the approximation of the cell boundaries by polygonal shapes, which is a requirement to use the shape function $\gamma_6$. Considering the full cellular boundaries and $q_6$, which is derived from the Minkowski tensors, leads to a different picture, with weaker hexatic order. A critical question in the literature has been whether shape measures for different *p*-atic orders can be directly compared. While some studies have suggested relationships between $\gamma_2$ and $\gamma_6$ (*Armengol-Collado et al., 2023*), our results refute this notion. We demonstrated that measures like $q_2$ and $q_6$ are independent and capture fundamentally distinct aspects of cell shape and alignment. Comparing them directly is mathematically but also physically and biologically misleading. We further tested the hypothesis of a hexatic-nematic crossover at larger length scales by coarse-graining $q_2$ and $q_6$. To discuss such a crossover requires direct comparison of $q_2$ and $q_6$ or their coarse-grain quantities $Q_2$ and $Q_6$, which is conceptually questionable. However, the results showed no consistent trends indicative of a crossover, regardless of the considered model or the experimental data. This leads to the conclusion that the proposed hexatic-nematic crossover in *Armengol-Collado et al., 2023* is not a physical phenomena but specific to the considered method.

Our findings suggest that *p*-atic orders should be studied independently, also across length scales, as they describe complementary aspects of cellular organization. The coexistence of distinct orientational orders emphasized in different studies—such as nematic ($p = 2$) (*Duclos et al., 2017*; *Saw et al., 2017*; *Kawaguchi et al., 2017*), tetratic ($p = 4$) (*Cislo et al., 2023*), and hexatic ($p = 6$) (*Li and Ciamarra, 2018*)—is not contradictory but highlights the rich, multifaceted nature of cellular organization. Rather than searching for a single dominant order, future research should focus on the interplay of different *p*-atic orders and their associated defects. This suggests to not only consider *p*-atic liquid crystal theories (*Giomi et al., 2022b*) for one specific *p*, but combinations of these models for various *p*'s. Understanding how these orders interact may reveal how they collectively regulate morphogenetic processes.

Connecting *p*-atic orders to biological function remains a critical avenue for exploration. While the mathematical independence of $q_2$, $q_6$, and other shape measures precludes the identification of a universal dominant order, biological systems may exhibit context-dependent preferences. For example, a specific *p*-atic order might correlate with or drive a key morphogenetic event. Investigating these connections could yield insights into how tissues achieve functional organization and adapt to environmental cues. As such, while it is difficult to speak of dominating orders from a mathematical point of view, there could be a dominating order from a biological point of view, meaning the *p*-atic order connected to the governing biological process.

## Acknowledgements

We acknowledge fruitful discussions with Björn Böttcher, Brendan Tobin, and Emma Happel. HJ acknowledges funding by the European Union's Horizon 2020 research and innovation programme under the Marie Skłodowska-Curie grant agreement No. 945371. RS acknowledges support from the UK Engineering and Physical Sciences Research Council (Award EP/W023946/1). AD acknowledges funding from the Novo Nordisk Foundation (grant No. NNF18SA0035142 and NERD grant No. NNF21OC0068687), Villum Fonden (grant No. 29476), and the European Union (ERC, PhysCoMeT, 101041418). AV acknowledges funding from the German Research Foundation (Award FOR3013 'Vector- and tensor-valued surface PDEs') and computing resources provided by JSC through MORPH and by ZIH through WIR. Views and opinions expressed are, however, those of the authors only and

do not necessarily reflect those of the European Union or the European Research Council. Neither the European Union nor the granting authority can be held responsible for them.

## Additional information

### Funding

| Funder | Grant reference number | Author |
|---|---|---|
| Deutsche Forschungsgemeinschaft | FOR3013 | Axel Voigt |
| Engineering and Physical Sciences Research Council | EP/W023946/1 | Rastko Sknepnek |
| Novo Nordisk Fonden | NNF18SA0035142 | Amin Doostmohammadi |
| European Research Council | 101041418 | Amin Doostmohammadi |

The funders had no role in study design, data collection and interpretation, or the decision to submit the work for publication.

### Author contributions

Lea Happel, Software, Formal analysis, Validation, Investigation, Visualization, Writing – original draft; Griseldis Oberschelp, Software, Formal analysis, Validation, Visualization, Methodology, Writing – original draft; Valeriia Grudtsyna, Data curation, Investigation; Harish P Jain, Software, Formal analysis; Rastko Sknepnek, Data curation, Software, Funding acquisition; Amin Doostmohammadi, Funding acquisition, Investigation, Methodology, Writing – review and editing; Axel Voigt, Conceptualization, Supervision, Funding acquisition, Investigation, Methodology, Writing – original draft, Project administration, Writing – review and editing

### Author ORCIDs

Lea Happel ![ORCID] https://orcid.org/0000-0002-4525-9185
Griseldis Oberschelp ![ORCID] https://orcid.org/0009-0000-1735-7493
Rastko Sknepnek ![ORCID] https://orcid.org/0000-0002-0144-9921
Amin Doostmohammadi ![ORCID] https://orcid.org/0000-0002-1116-4268
Axel Voigt ![ORCID] https://orcid.org/0000-0003-2564-3697

Reviewer #1 (Public review): https://doi.org/10.7554/eLife.105680.3.sa1
Reviewer #3 (Public review): https://doi.org/10.7554/eLife.105680.3.sa2
Author response https://doi.org/10.7554/eLife.105680.3.sa3

## Additional files

### Supplementary files
MDAR checklist

### Data availability

A code illustrating the extraction of the contour and the calculation from $q_p$ and $\vartheta_p$ for grayscale images can be found on Zenodo at https://doi.org/10.5281/zenodo.15430268. Simulation code for the vertex and multiphase field models can be found at https://github.com/sknepneklab/RheoVM (*Sknepnek, 2023*) and https://gitlab.mn.tu-dresden.de/iwr/amdis (*Vey et al., 2022*), respectively. The data from *Armengol-Collado et al., 2023* to which we compare in the result section was made publicly available by the authors of said paper under GitHub at https://github.com/hexanematic/orientation_tracker.

The following dataset was generated:

| Author(s) | Year | Dataset title | Dataset URL | Database and Identifier |
|---|---|---|---|---|
| Happel L, Oberschelp G, Tobin B | 2025 | Quantifying the shape of cells - from Minkowski tensors to p-atic orders | https://doi.org/10.5281/zenodo.15430268 | Zenodo, 10.5281/zenodo.15430268 |

The following previously published dataset was used:

| Author(s) | Year | Dataset title | Dataset URL | Database and Identifier |
|---|---|---|---|---|
| Armengol-Collado J-M, Carenza LN, Eckert J, Krommydas D, Giomi L | 2025 | Orientation tracker | https://github.com/hexanematic/orientation_tracker | GitHub, hexanematic/orientation_tracker |

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

## Appendix 1

### Simulation methods

We consider two modeling approaches, an active vertex model and a multiphase field model. Both have been proven to capture various generic properties of epithelial tissue (*Hakim and Silberzan, 2017*; *Alert and Trepat, 2020*; *Moure and Gomez, 2021*). The key aspects of formulations are outlined in the sections Appendix 1 Active vertex model and Appendix 1 Multiphase field model. They refer to *Koride et al., 2018*; *Maroudas-Sacks et al., 2021*; *Killeen et al., 2022* and (*Wenzel and Voigt, 2021*; *Jain et al., 2023*; *Jain et al., 2024*), respectively. Parameters used within these models are listed in *Appendix 1—table 1* and *Appendix 1—table 2*.

The initial configuration for the active vertex model simulations was built by placing $N$ points randomly in a periodic simulation box of length $L \times L$ and ensuring that no two points were within a distance less than $r_{\mathrm{cut}}$ from each other. The points were used to create a Voronoi tiling. Typically, such a tiling had cells of very irregular shapes. To make cell shapes more uniform, the centroids of each tile were computed and used as seeds for a new Voronoi tiling. This process was iterated until it converged within the tolerance of $10^{-5}$, resulting in a so-called well-centered Voronoi tiling with cells of random shapes but similar sizes. Initial directions of polarity vectors $\hat{\mathbf{n}}_{C_i}$ were chosen at random from a uniform distribution. The initial configuration for the multiphase field model simulations considers a regular arrangement of $N$ cells of equal size in a periodic simulation box of length $L \times L$ with randomly chosen directions of self-propulsion and simulating for several time steps until the cell shapes appear sufficiently irregular. For both models, we consider 100 cells and analyse time instances of the evolution.

Active vertex model

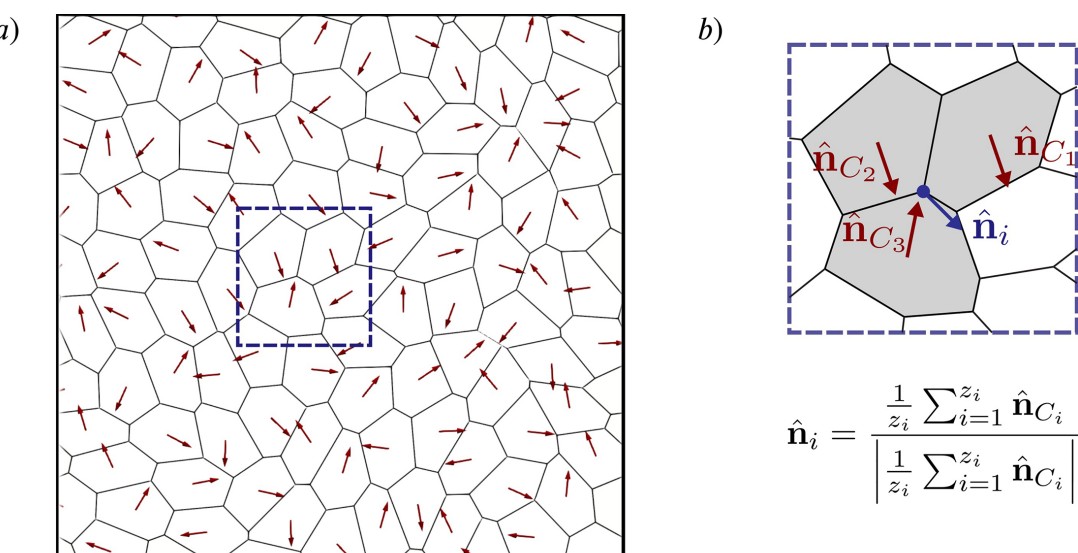

**Appendix 1—figure 1.** Illustration of the active vertex model. (**a**) Cell contour of the active vertex model. Red arrows represent the polarity vectors that set each cell's instantaneous direction of self-propulsion. (**b**) Zoom in on a vertex surrounded by three cells showing how the direction of self-propulsion on a vertex is calculated.

The active vertex model is based on models discussed in *Koride et al., 2018*; *Maroudas-Sacks et al., 2021*; *Killeen et al., 2022*. A confluent epithelial tissue is appreciated as a two-dimensional tiling of a plane with the elastic energy given as (*Honda, 1983*; *Farhadifar et al., 2007*; *Honda, 2022*)

$$E_{\mathrm{VM}} = \frac{K_A}{2} \sum_C \left( A_C - A_0 \right)^2 + \frac{K_P}{2} \sum_C \left( P_C - P_0 \right)^2, \tag{13}$$

where $A_C$ and $P_C$ are, respectively, area and perimeter of cell $C$, $A_0$ and $P_0$ are, respectively, preferred area and perimeter, and $K_A$ and $K_P$ are, respectively, area and perimeter moduli. For simplicity, it is assumed that all cells have the same values of $A_0$, $P_0$, $K_A$, and $K_P$. The model can be made dimensionless by dividing **Equation 13** by $e^* = K_A A_0^2$,

$$e_{\text{VM}} = \frac{1}{2} \sum_C (a_C - 1)^2 + \frac{k_P}{2} \sum_C (p_C - p_0)^2, \tag{14}$$

where $a_C = A_C/A_0$ (i.e. $l^* = \sqrt{A_0}$ is the unit of length), $p_c = P_C/\sqrt{A_0}$, $k_P = K_P/(K_A A_0)$, and $p_0 = P_0/\sqrt{A_0}$ is called the shape index, and it has been shown to play a central role in determining if the tissue is in a solid or fluid state (**Bi et al., 2015**).

One can use **Equation 13** to find the mechanical force on a vertex $i$ as $\mathbf{F}_i = -\nabla_{\mathbf{r}_i} E_{\text{VM}}$, where $\nabla_{\mathbf{r}_i}$ is the gradient with respect to the position $\mathbf{r}_i$ of the vertex. The expression for $\mathbf{F}_i$ is given in terms of the position of the vertex $i$ and its immediate neighbours (**Tong et al., 2023**), which makes it fast to compute. Furthermore, cells move on a substrate by being noisily self-propelled along the direction of their planar polarity described by a unit-length vector $\hat{\mathbf{n}}_{C_i}$ from which the polarity direction at the vertex $\hat{\mathbf{n}}_i$ is defined, see **Appendix 1—figure 1** b, where $z_i$ is the number of neighbours of vertex $i$. It is convenient to write $\hat{\mathbf{n}}_i = \cos(\alpha_i)\mathbf{e}_x + \sin(\alpha_i)\mathbf{e}_y$, where $\alpha_i$ is an angle with the x-axis of the simulation box. The equations of motion for vertex $i$ is a force balance between active, elastic, and frictional forces and are given as

$$\dot{\mathbf{r}}_i = v_0 \hat{\mathbf{n}}_i + \frac{1}{\gamma_{fr}} \mathbf{F}_i, \quad \dot{\alpha}_i = \xi_i(t), \tag{15}$$

where $v_0$ is the magnitude of the self-propulsion velocity defined as $f_0/\gamma_{fr}$. Furthermore, the overdot denotes the time derivative, $\gamma_{\text{fr}}$ is the friction coefficient, $f_0$ is the magnitude of the self-propulsion force (i.e. the activity), $\xi_i(t)$ is Gaussian noise with $\langle \xi_i(t) \rangle = 0$ and $\langle \xi_i(t)\xi_j(t') \rangle = 2D_r \delta_{ij} \delta(t - t')$, where $D_r$ is the rotation diffusion constant and $\langle \cdot \rangle$ is the ensemble average over the noise. Equations of motion can be non-dimensionalized by measuring time in units of $t^* = \gamma_{\text{fr}}/(K_A A_0)$ and force in units of $f^* = K_A A_0^{3/2}$ and integrated numerically using the first-order Euler-Maruyama method with timestep $\tau$.

## Multiphase field model

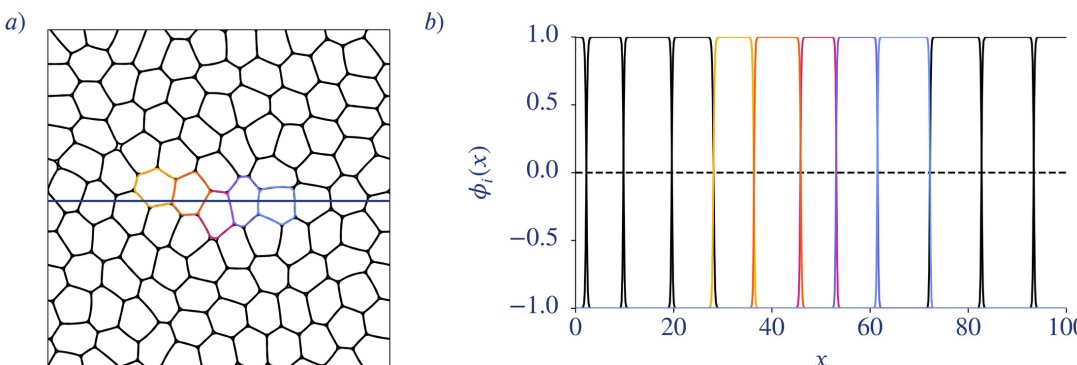

**Appendix 1—figure 2.** Illustration of the multiphase field model. (**a**) Cell contours of the multiphase field model. (**b**) Corresponding phase field functions along the horizontal line in (**a**). Colours correspond to the ones in (**a**).

The multiphase field modeling approach follows **Wenzel and Voigt, 2021**; **Jain et al., 2023**; **Jain et al., 2024**. Each cell is described by a scalar phase field variable $\phi_i$ with $i = 1, 2, \ldots, N$, where the bulk values $\phi_i \approx 1$ denotes the cell interior, $\phi_i \approx -1$ denotes the cell exterior, and with a diffuse interface of width $\mathcal{O}(\epsilon)$ between them representing the cell boundary. The phase field $\phi_i$ follows the conservative dynamics

$$\partial_t \phi_i + \mathbf{v}_i \cdot \nabla \phi_i = \Delta \frac{\delta \mathcal{F}}{\delta \phi_i}, \tag{16}$$

with the free energy functional $\mathcal{F} = \mathcal{F}_{CH} + \mathcal{F}_{INT}$ containing the Cahn-Hilliard energy

$$\mathcal{F}_{CH} = \frac{1}{Ca} \sum_{i=1}^{N} \int_{\Omega} \left( \frac{\epsilon}{2} \|\nabla \phi_i\|^2 + \frac{1}{4\epsilon} (\phi_i^2 - 1)^2 \right) d\mathbf{x}, \tag{17}$$

where the Capillary number $Ca$ is a parameter for tuning the cell deformability, and an interaction energy consisting of repulsive and attractive parts defined as

$$\mathcal{F}_{INT} = \frac{1}{In} \sum_{i=1}^{N} \int_{\Omega} \sum_{j \neq i} a_r (\phi_i + 1)^2 (\phi_j + 1)^2 - a_a (\phi_i^2 - 1)^2 (\phi_j^2 - 1)^2 d\mathbf{x}, \tag{18}$$

where $In$ denotes the interaction strength, and $a_r$ and $a_a$ are parameters to tune contribution of repulsion and attraction. The repulsive term penalises overlap of cell interiors, while the attractive part promotes overlap of cell interfaces. The relation to former formulations is discussed in *Happel and Voigt, 2024*.

Each cell is self-propelled, and the cell activity is introduced through an advection term. The cell velocity field is defined as

$$\mathbf{v}_i(\mathbf{x}, t) = v_0 \mathbf{e}_i(t) \hat{\phi}_i(\mathbf{x}, t), \tag{19}$$

where $v_0$ is used to tune the magnitude of activity, $\hat{\phi}_i = \frac{\phi_i + 1}{2}$ and $\mathbf{e}_i = [\cos \theta_i(t), \sin \theta_i(t)]$ is its direction. The migration orientation $\theta_i(t)$ evolves diffusively with a drift that aligns to the principal axis of cell's elongation as $d\theta_i = \sqrt{2D_r} dW_i(t) + \alpha(\beta_i(t) - \theta_i(t)) dt$, where $D_r$ is the rotational diffusivity, $W_i$ is the Wiener process, $\beta_i(t)$ is the orientation of the cell elongation and $\alpha$ controls the time scale of this alignment.

The resulting system of partial differential equations is considered on a square domain with periodic boundary conditions and is solved by the finite element method within the toolbox AMDiS (*Vey and Voigt, 2007*; *Witkowski et al., 2015*) and the parallelization concept introduced in *Praetorius and Voigt, 2018* is considered, which allows scaling with the number of cells. We, in addition, introduce a de Gennes factor in $\mathcal{F}_{CH}$ to ensure $\phi_i \in [-1, 1]$, see *Salvalaglio et al., 2021*.

## Parameters for the computational models

**Appendix 1—table 1.** Values of the dimensionless parameters used in the active vertex model.

| Parameter | Description | Numerical value |
|---|---|---|
| $N$ | Number of cells | 100 |
| $L$ | Simulation box size | 10 |
| $T$ | Total simulation time | 300 |
| $k_P$ | Perimeter elastic modulus | 1.0 |
| $\tau$ | Simulation time step | 0.01 |
| $v_0$ | Self-propulsion strength (i.e. activity) | 0.1–0.4 |
| $D_r$ | Rotation diffusion coefficient | 0.05 |
| $p_0$ | Shape index of active cells | 3.5–3.875 |

**Appendix 1—table 2.** Values of the dimensionless parameters used in the multiphase field model.

| Parameter | Description | Numerical value |
|---|---|---|
| $N$ | Number of cells | 100 |
| $L$ | Simulation box size | 100 |

*Appendix 1—table 2 Continued on next page*

*Appendix 1—table 2 Continued*

| Parameter | Description | Numerical value |
|---|---|---|
| $T$ | Total simulation time | 150 |
| $\epsilon$ | Interface width | 0.15 |
| $\tau$ | Simulation time step | 0.005 |
| $v_0$ | Self-propulsion strength (i.e. activity) | 0.4–1.0 |
| $D_r$ | Rotation diffusion coefficient | 0.01 |
| $\alpha$ | Alignment parameter | 0.1 |
| $Ca$ | Capillary number | 0.05–0.2 |
| $In$ | Interaction number | 0.1 |
| $a_a$ | Cell-cell attraction strength | 1.0 |
| $a_r$ | Cell-cell repulsion strength | 1.0 |
| $h$ | Mesh size | $5h \approx \epsilon$ |

## Experimental setup

### Cell culture

Madin-Darby canine kidney (MDCK) cells were cultured in DMEM (DMEM, low glucose, GlutaMAX Supplement, pyruvate) supplemented with 10 % fetal bovine serum (FBS; Gibco) and 100 U/mL penicillin/streptomycin (Gibco) at 37 °C with 5% $CO_2$. The cell line was tested for mycoplasma.

### Monolayer preparation

Cells were seeded on glass-bottom dishes (Mattek) pretreated with 10 μg/mL fibronectin (human plasma; Gibco) in phosphate-buffered saline (PBS, pH 7.4; Gibco). Fibronectin was incubated for 30 min at 37 °C. The initial cell seeding density was sparse. The sample was imaged approximately 24 hr later, when a confluent monolayer had formed.

### Live cell imaging

The sample was imaged using a Nikon ECLIPSE Ti microscope equipped with an H201-K-FRAME chamber, heating system (Okolab), and $CO_2$ pump (Okolab), which maintained environmental conditions at 37°C and 5% $CO_2$. Phase-contrast images were acquired using a 10×, NA = 0.3 Plan Fluor objective and an Andor Neo 5.5 sCMOS camera.

### External dataset

As a complementary approach, we analyzed data from the study by Armengol-Collado et al. *Armengol-Collado et al., 2023*, which included images of MDCK GII monolayers labeled for E-cadherin. These images were made publicly available by the authors via GitHub.

### Image analysis

All image data, including both our phase-contrast recordings and the E-cadherin-labeled external dataset, were analyzed using Cellpose *Stringer et al., 2021*. Segmentation was performed using manually trained models, each tailored to its respective dataset.

### Extraction of the contour

Starting out from the segmented images (stored as grayscale images), we extract the contour of the cells using the Python package *scikit-image* (*van der Walt et al., 2014*). To get rid of pixel-shaped artifacts, we smooth the contour by replacing every coordinate by the average over nine neighboring points in the outline.

## Appendix 2

### Results for $q_3$, $q_4$, and $q_5$
Distribution functions in dependence of deformability

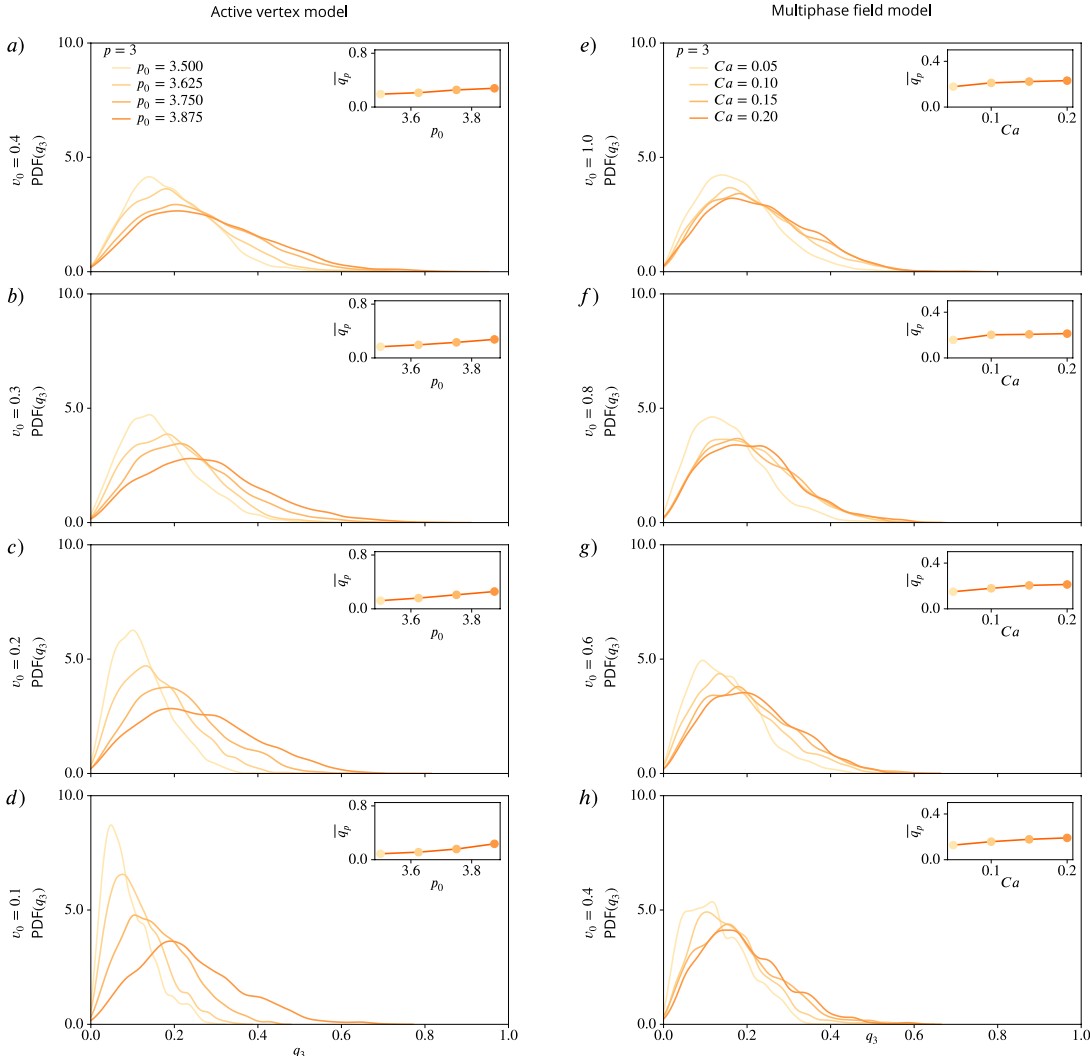

**Appendix 2—figure 1.** PDFs for $q_3$ using kde-plots, for varying deformability $p_0$ or $Ca$ and fixed activity $v_0$. Inlets show mean values of $q_3$ as function of deformability. (**a-d**) Active vertex model, (**e-h**) Multiphase field model for decreasing activity.

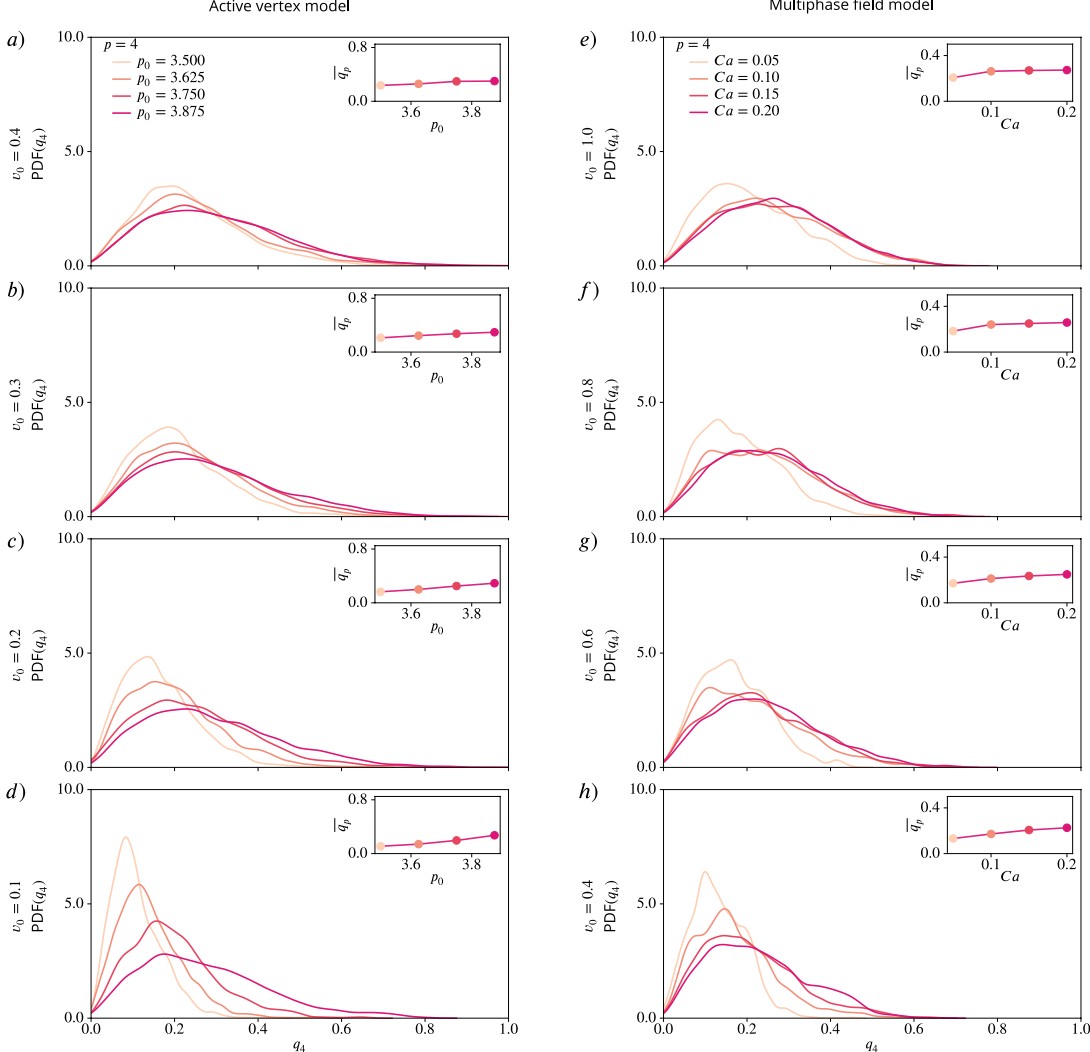

**Appendix 2—figure 2.** PDFs for $q_4$ using kde-plots, for varying deformability $p_0$ or $Ca$ and fixed activity $v_0$. Inlets show mean values of $q_4$ as function of deformability. (**a-d**) Active vertex model, (**e-h**) Multiphase field model for decreasing activity.

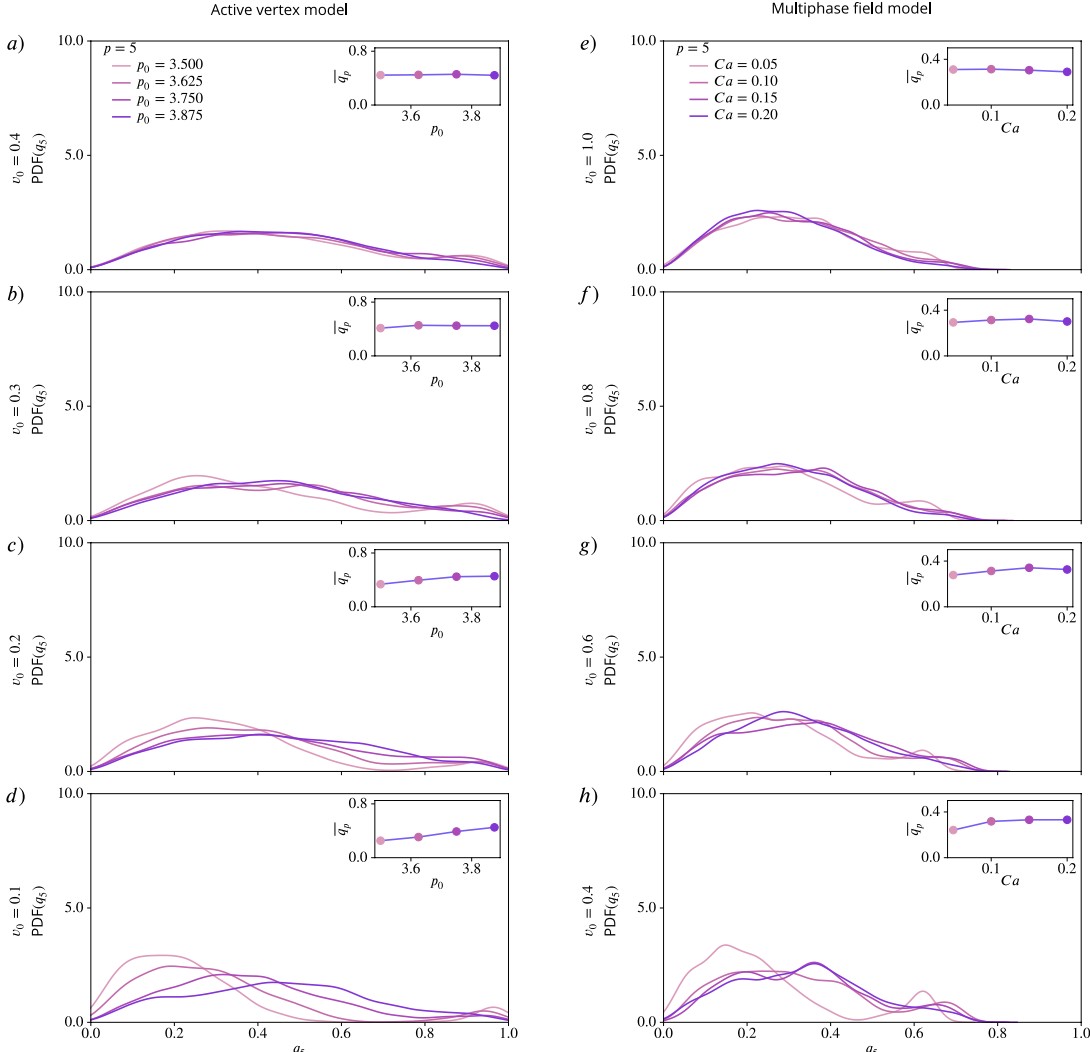

**Appendix 2—figure 3.** PDFs for $q_5$ using kde-plots, for varying deformability $p_0$ or $Ca$ and fixed activity $v_0$. Inlets show mean values of $q_5$ as function of deformability. (**a-d**) Active vertex model, (**e-h**) Multiphase field model for decreasing activity.

# Distribution functions in dependence of activity

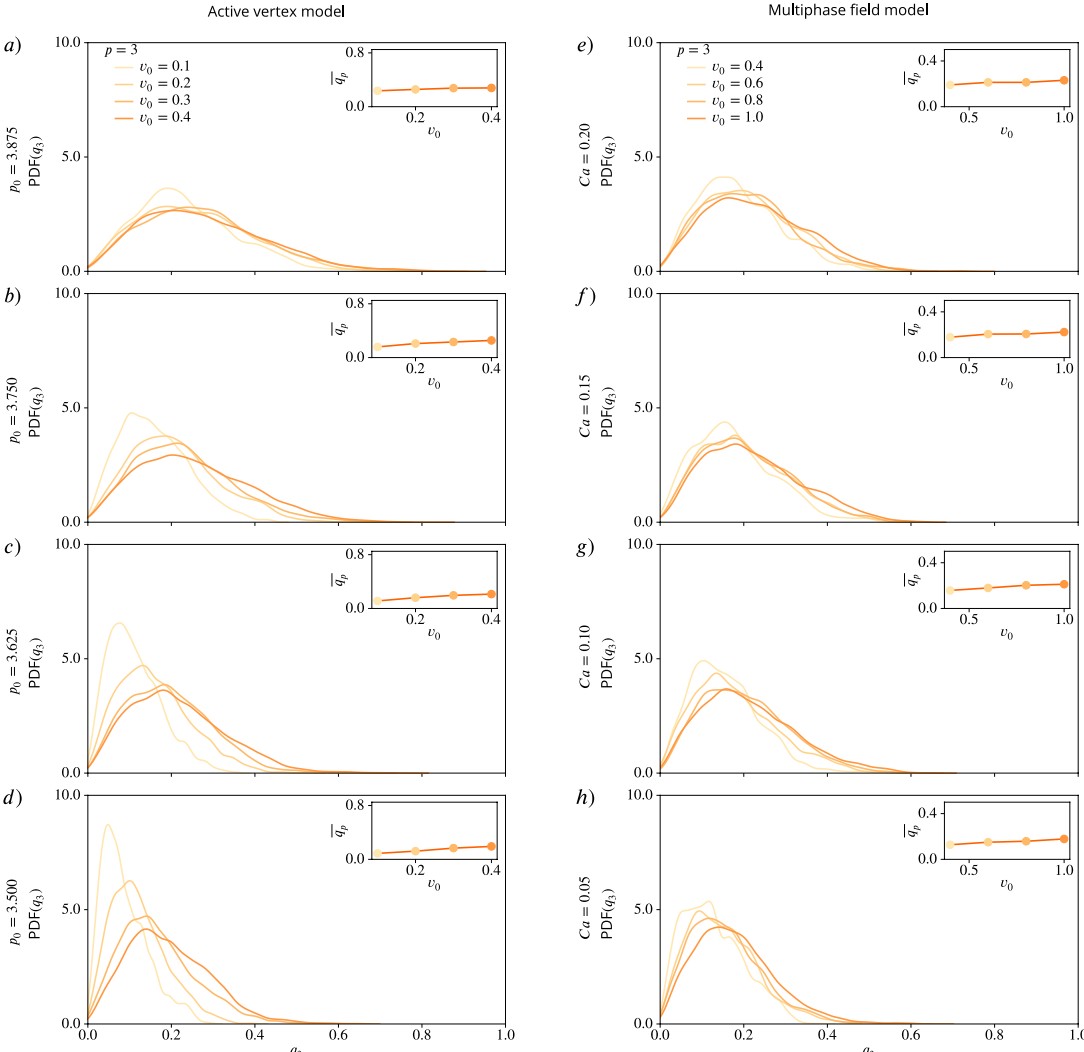

**Appendix 2—figure 4.** PDFs for $q_3$ using kde-plots, for varying activity $v_0$ and fixed deformability $p_0$ or $Ca$. Inlets show mean values of $q_3$ as function of activity. (**a-d**) Active vertex model and (**e-h**) Multiphase field model for decreasing deformability.

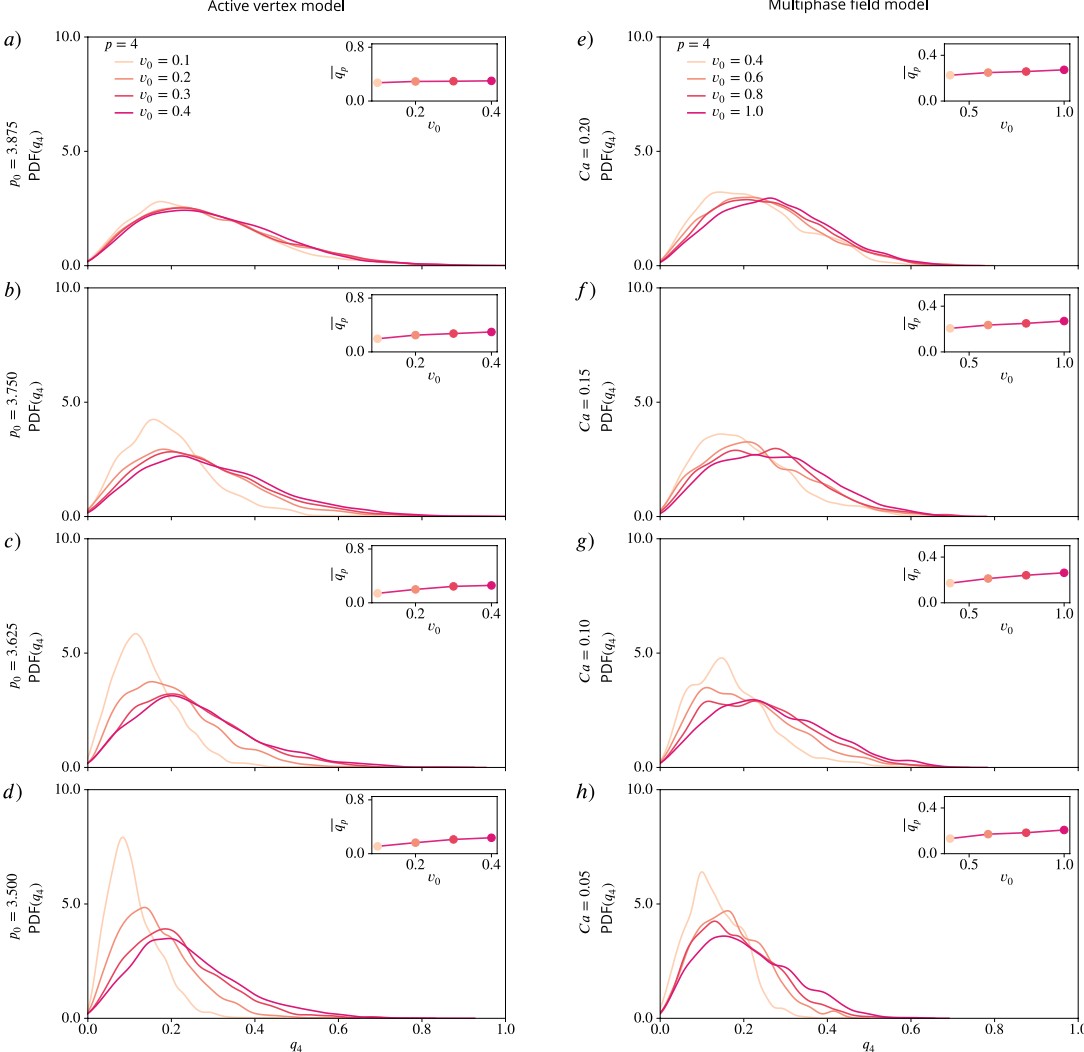

**Appendix 2—figure 5.** PDFs for $q_4$ using kde-plots, for varying activity $v_0$ and fixed deformability $p_0$ or $Ca$. Inlets show mean values of $q_4$ as function of activity. (**a-d**) Active vertex model and (**e-h**) Multiphase field model for decreasing deformability.

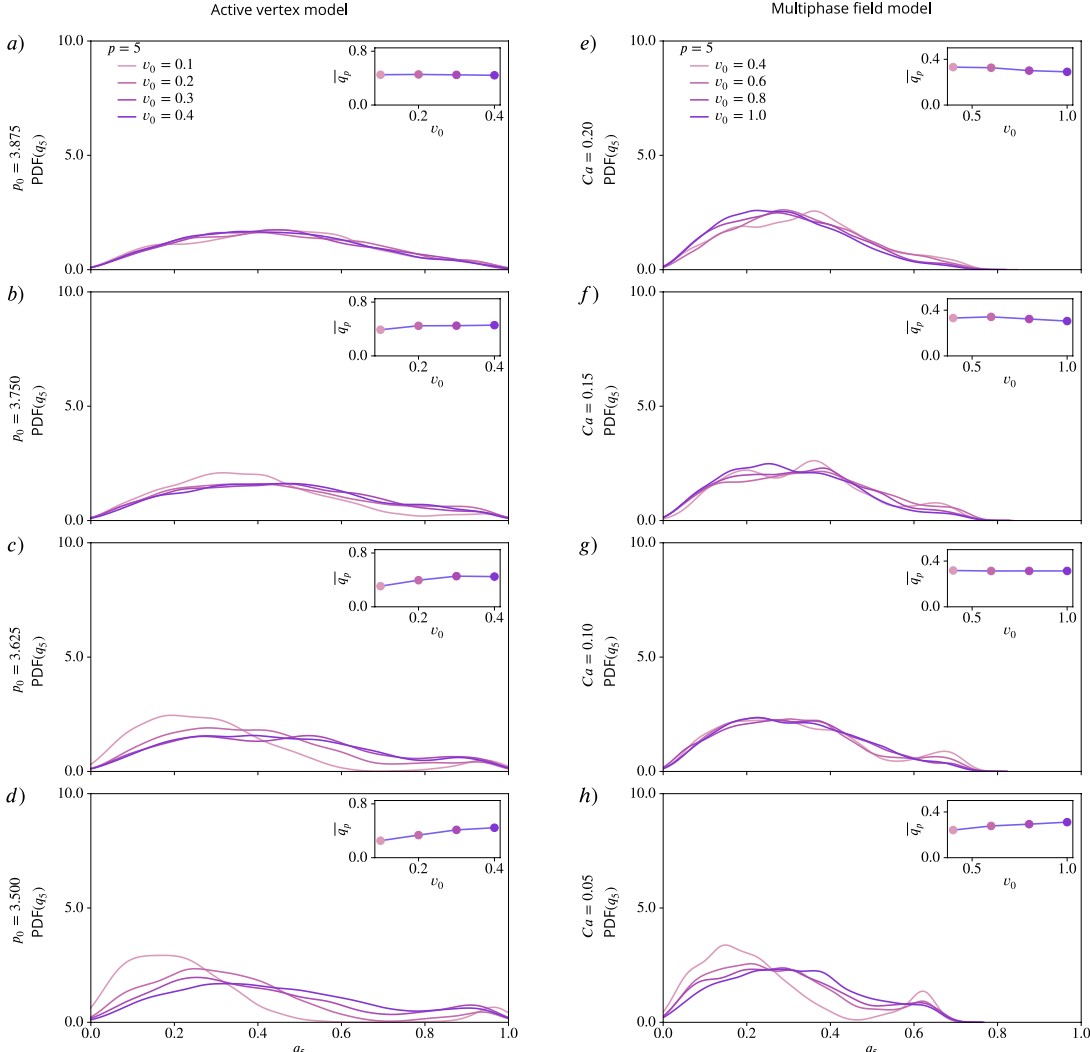

**Appendix 2—figure 6.** PDFs for $q_5$ using kde-plots, for varying activity $v_0$ and fixed deformability $p_0$ or $Ca$. Inlets show mean values of $q_5$ as function of activity. (**a-d**) Active vertex model and (**e-h**) Multiphase field model for decreasing deformability.

## Summarized behavior in dependence of activity and deformability

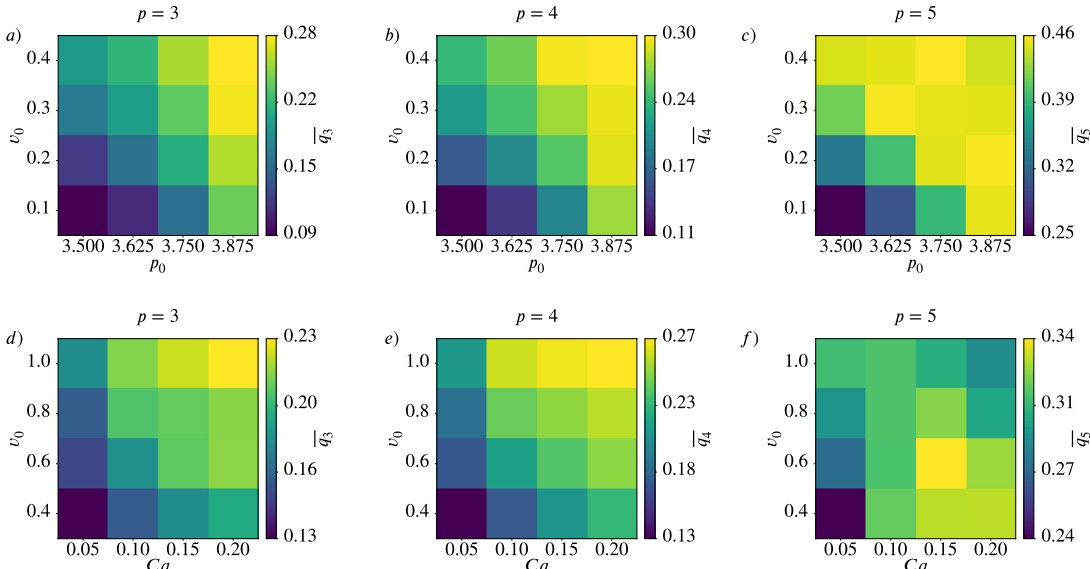

**Appendix 2—figure 7.** Mean value $\overline{q_p}$ for $p = 3$ (left), $p = 4$ (middle) and $p = 5$ (right) as a function of deformability $p_0$ or $Ca$ and activity $v_0$ for active vertex model (**a-c**) and multiphase field model (**d-f**).

## Results using polygonal shape analysis

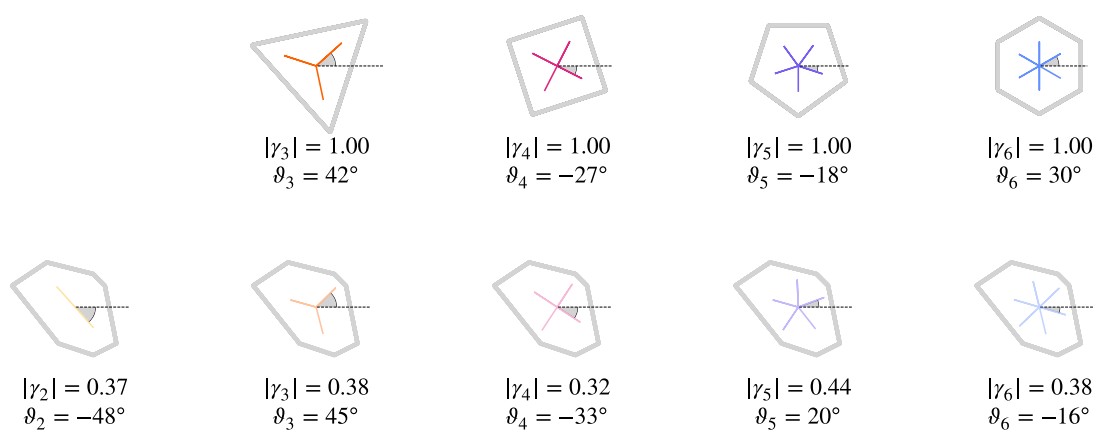

**Appendix 2—figure 8.** Regular and irregular shapes, adapted from ***Armengol-Collado et al., 2023***, with magnitude and orientation calculated by ***Equation 8*** and ***Equation 9***. The brightness scales with the magnitude $|\gamma_p|$.

To visualize the similarities and differences between $q_P$ and $\gamma_p$ we show the analogue of ***Figure 4*** using the Minkowski tensors in ***Appendix 2—figure 8*** using polygonal shape analysis. For regular polygons, both shape characterization methods behave in the same way as seen in the first row. For irregular shapes, like in the second row, they already show a different behaviour. Note that the rounded shapes from ***Figure 4*** are missing in ***Appendix 2—figure 8*** as $\gamma_p$ requires a polygon.

The influence of the choice of the shape characterization method is not only visible in the values for single shapes, it can also be seen in the mean and standard deviation. To illustrate this, we use $\gamma_p$ to characterize the shapes from the same experimental data as in ***Figures 1 and 2***. The segmented data consists of smooth contours for each cell. As $\gamma_p$ only works with vertex coordinates, the first step is to identify these. Note that this always leads to the approximation of the cell shape with a polygon and, therefore to a simplification of the shape. For finding the vertices, we consider the Voronoi interface method (***Saye and Sethian, 2011***; ***Saye and Sethian, 2012***). Using the so obtained

vertices and *Equation 8* and *Equation 9* leads to the results shown in *Appendix 2—figures 9 and 10*. Compared to *Figures 1 and 2*, where the Minkowski tensor was used for the same data, we see that the choice of the shape characterization method highly influences the results.

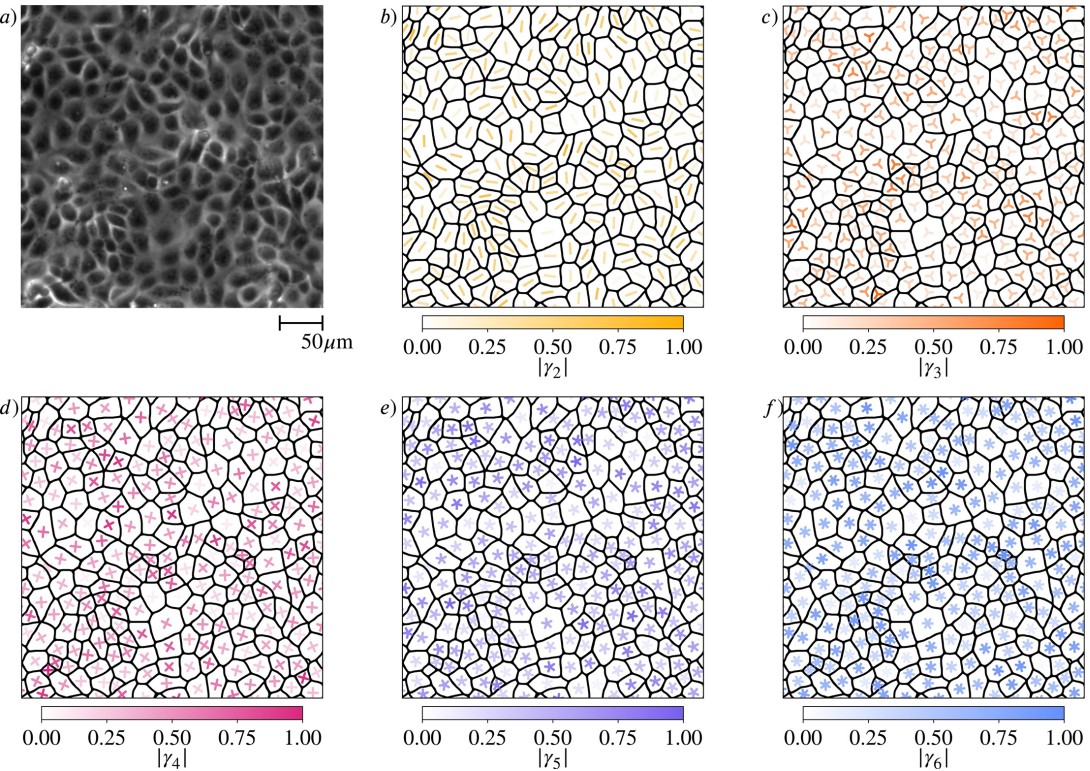

**Appendix 2—figure 9.** Shape classification of cells in wild-type Madin-Darby canine kidney (MDCK) cell monolayer. (**a**) Raw experimental data. (**b-f**) Polygonal shape classification, visualized using $\gamma_p$ calculated by *Equation 8* and *Equation 9* for $p = 2, 3, 4, 5, 6$, respectively. The brightness and the rotation of the $p$-atic director indicates the magnitude and the orientation, respectively. See Appendix 1-Experimental setup for details on the experimental data.

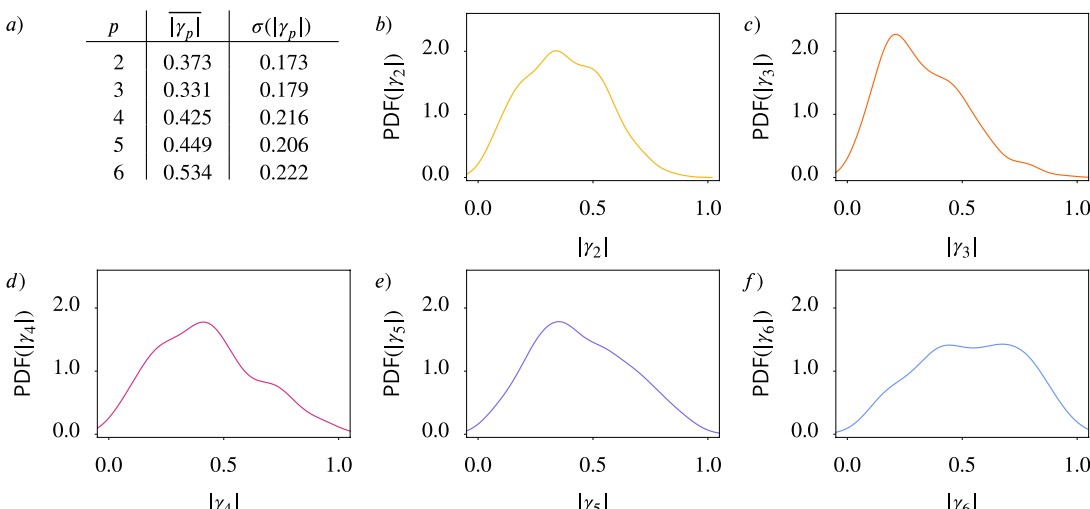

**Appendix 2—figure 10.** Statistical data for cell shapes identified in *Appendix 2—figure 9*. (**a**) Mean $\overline{|\gamma_p|}$ and standard deviation $\sigma(|\gamma_p|)$ of $|\gamma_p|$. (**b- f**) Probability distribution function (PDF) of $|\gamma_p|$ for $p = 2, 3, 4, 5, 6$, respectively. Kde-plots are used to show the probability distribution. For this first analysis, we regard only one frame with 235 cells.

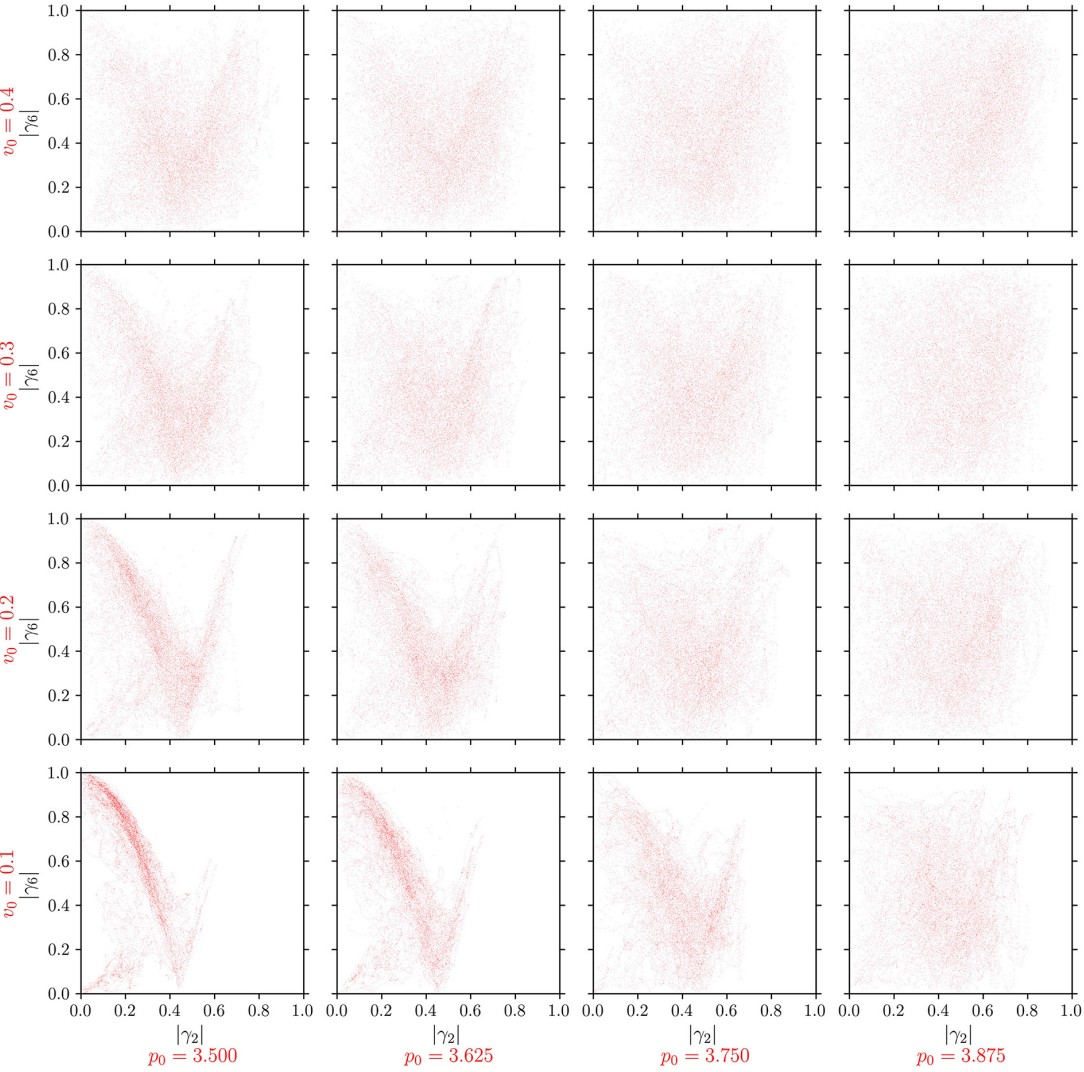

**Appendix 2—figure 11.** $|\gamma_6|$ (y-axis) versus $|\gamma_2|$ (x-axis) for all cells in the active vertex model. For each cell and each timestep, we plot one point ($|\gamma_2|$, $|\gamma_6|$). Each panel corresponds to specific model parameters $p_0$ and $v_0$, representing deformability and activity.

The online version of this article includes the following figure supplement(s) for appendix 2—figure 11:

**Appendix 2—Figure 11 supplement 1.** Distance correlation $dCor(|\gamma_2|, |\gamma_6|)$ for the simulation data of the active vertex model.

**Appendix 2—Figure 11 supplement 2.** P-values of the distance correlation $P_{dCor(|\gamma_2|,|\gamma_6|)}$ for the simulation data of the active vertex model.

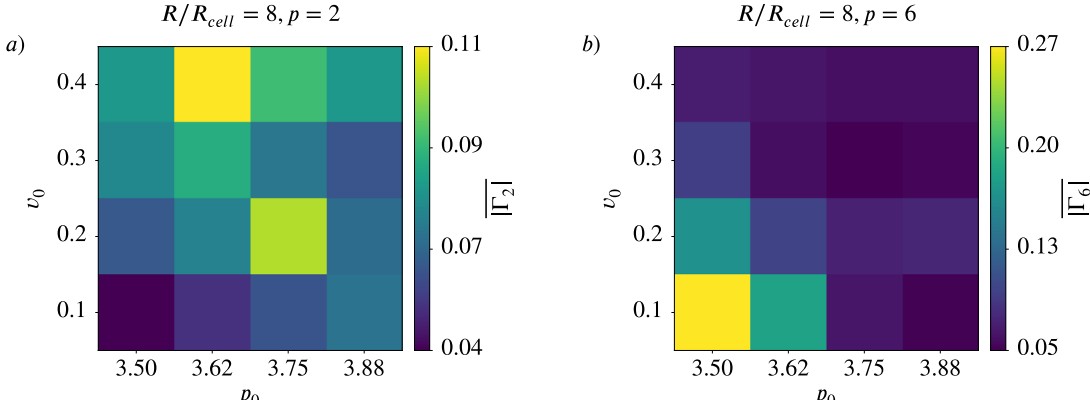

**Appendix 2—figure 12.** Coarse-Grained nematic ($p = 2$) and hexatic ($p = 6$) order for $R/R_{cell} = 8$ depending on activity and deformability of the cells. $\overline{|\Gamma_p|}$ as function of deformability $p_0$ and activity $v_0$ for active vertex model. (a) nematic order ($p = 2$), (b) hexatic order ($p = 6$).

The online version of this article includes the following figure supplement(s) for appendix 2—figure 12:

**Appendix 2—Figure 12 supplement 1.** $\overline{\Gamma_6}$ versus $\overline{\Gamma_2}$ for different coarse-graining radii in the active vertex model.

Qualitative consequences of the approach become apparent by comparing *Figure 8* with *Figure 12*, which, instead of the monotonic trend for activity and deformability, exhibit nonmonotonic trends, peaking at intermediate values of activity or deformability.

Considering the independence of $\gamma_2$ and $\gamma_6$ the corresponding results to *Figure 7* and the corresponding distance correlation and statistical tests *Figure 7—figure supplements 1 and 2*, respectively, are provided in *Appendix 2—figures 11* and *Appendix 2 Figure 11—figure Supplement 1*, *Appendix 2—figure 11—figure supplement 2* respectively, but only for the active vertex model.

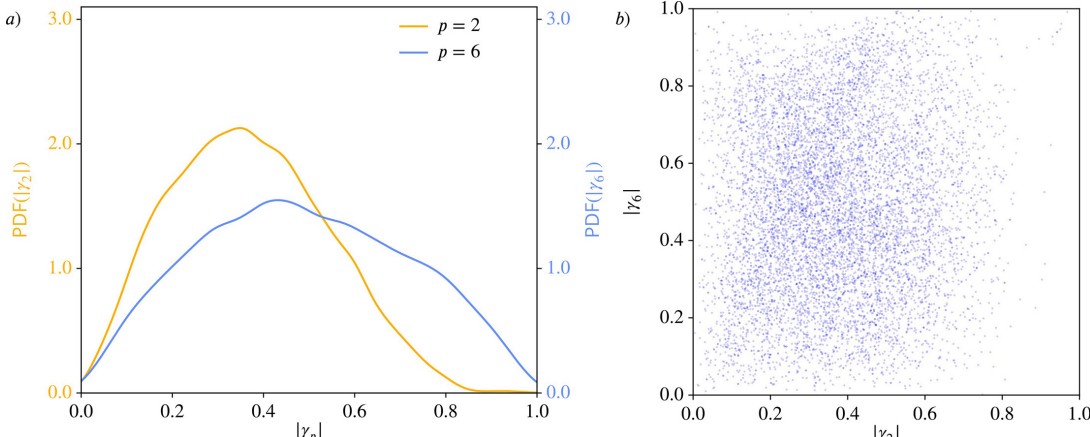

**Appendix 2—figure 13.** Nematic ($p = 2$) and hexatic ($p = 6$) order for the cells in the experiments from *Armengol-Collado et al., 2023*. We use the polygonal approximation of the cell shape as $\gamma_p$ can only work with polygons. (a): Probability distribution functions (PDFs) using kde-plots, for |$\gamma_2$| (orange) and |$\gamma_6$| (blue). (b): Nematic ($p = 2$) and hexatic ($p = 6$) order are independent of eachother. |$\gamma_6$| (y-axis) versus |$\gamma_2$| (x-axis) for all cells. For each cell and each timestep, we plot one point (|$\gamma_2$|, |$\gamma_6$|).

The online version of this article includes the following figure supplement(s) for appendix 2—figure 13:

**Appendix 2—Figure 13 supplement 1.** Distance correlation $dCor(|\gamma_2|,|\gamma_6|)$ for all cells from the experimental data in *Armengol-Collado et al., 2023*.

**Appendix 2—Figure 13 supplement 2.** P-value of the distance correlation $P_{dCor(|\gamma_2|,|\gamma_6|)}$ for all cells from the experimental data in *Armengol-Collado et al., 2023*.

