## [Editor Report · eLife Assessment]

This **important** work describes a set of parameters that give a robust description of shape features of cells in tissues. The evidence for the usefulness of these parameters is **solid**. The work should be of interest for anybody analyzing epithelial dynamics, but more details about the analysis of experimental images are necessary and some streamlining of the text would increase the accessibility of the material for non-specialists.

---

## [Referee Report · Reviewer #1 (Public review)]

Summary:

The authors stated aim is to introduce so-called Minkowski tensors to characterize and quantify the shape of cells in tissues. The authors introduce Minkowski tensors and then define the p-atic order q_p as a cell shape measure, where p is an integer. They also introduce a previously defined measure of p-atic order in the form of the parameter \gamma_p. The authors compute q_p for data obtained by simulating an active vertex model and a multiphase field model, where they focus on p=2 and p=6 - so-called nematic and hexatic order - as the two values of highest biological relevance. Based on their analysis, the authors state that q_2 and q_6 are independent, that there is no crossover for the coarse-grained quantities, that a comparison of q_p for different values of p is not meaningful, and determine the dependence of the mean value of q_2 and q_6 on cell activity and deformability. Subsequently, they apply their method to data from MDCK monolayers and argue that the full range of q_p values needs to be considered to characterize shape and positional order in epithelia..

Strength:

The work presents a set of parameters that are useful for analyzing cell shape.

Weaknesses:

The introduction of the Minkowski tensors is hardly accessible for typical biologists. Eventually, most quantification is done using q_p, which can be defined without recursion to Minkowski functionals. The relation to Minkowski functionals makes the important properties of robustness and stability evident. However, for an audience of biologists, the derivation of this property could be relegated to an Appendix. Instead, the text could directly go to the results of the analysis of experimental and modeling data.

Important details about how the cell shapes are extracted from the experimental data are missing. The two data sets the authors consider are not analyzed in the same way.

---

## [Referee Report · Reviewer #3 (Public review)]

Hapel et al. present an article entitled Quantifying the shape of cells - from Minkowski tensors to p-atic order. The paper reports the p-atic quantitative method - established in physics - to extract cell full shapes in biological experiments using their images of epithelial MDCK cells (phase contrast) and also images reported in another paper as well as their own simulations based on active vertex model and multiphase phase fields approaches. Authors present the rationale of this new strategy for quantification. They adapt the method of Minkowski tensors and they extract distributions of cell shapes readouts with plots of their distributions. An emphasis is given to changes in cell shapes captured by this method. Higher rank tensors are considered as well as representations with intuitive meanings and q_i orders and their potential correlations or absence of correlations - for example q_2 and q_6, leading to statements about nematic and hexatic orders.

This analysis and its strength are contrasted with Armengol-Collade et al. (2023) quoted in the paper, who consider polygonal shapes for cells and their shape function 𝛾_p. Authors support the notion of a key improvement thanks to Minkowski tensors approach and doing so, they challenge the former crossovers correlations statements reported in Armengol-Collade et al. (2023). In this context, they defend that nematic liquid crystals approach is not sufficient to capture cell dynamics in tissues. Also they propose that q_2 and q_6 could serve as readout for activity and deformability of cells among other statements related to their approach.

A variety of analytical methods have been realised to track cells in monolayers in vitro and in vivo during morphogenesis - for example, shear decomposition (from MPI-PKS Dresden) or links joining centroids and their neighbours approach (MSC/Curie Paris) to name few examples. It will be interesting in the future that systematic comparisons between these analytical methods are performed with highlights on their respective advantages and drawbacks. This will allow experimentalists to identify the best relevant methods to address their morphogenetic questions.

---

## [Author Response]

The following is the authors’ response to the original reviews

**Reviewer 1:**
I would suggest that the authors focus on what I think is the main goal of the work, namely, to consider the whole cell contour when characterizing cell shape instead of only some points on the contour. A reference to the connection with Minkowski tensors and the biologically relevant mathematical consequences of this connection would suffice; a detailed definition of the Minkowski tensors does not seem to be necessary. Especially because you do not really use them. You could use the analysis of the simulation data to explain what the *γp* miss and for which statements they would be sufficient.

We argue that the explanation of Minkowski tensors is helpful and should remain in the Methods and materials section. There are two reasons: First, our argumentation relays on the robustness and stability properties of Minkowski tensors. Introducing *qp* without the connection to Minkowski tensors would not allow us to make these statements. Second, Minkowski tensors seem not well known in the community, otherwise measures like *γp* would not have been introduced. Furthermore, readers not interested in the technical details could skip this part of the manuscript and directly go to the Results section. Concerning the questions, what the *γp* miss and for which statements they would be sufficient, the answer from a purly mathematical point of view is rather simple: As *γp* does not share robustness and stability it should not be used in any case! The provided results on computational and experimental data demonstrate the consequences of using such measures. In case of the proposed nematic-hexatic transition in Armengol-Collade et al. (2023) the consequence is severe, as this transition is specific only to the used method but not to the underlying physics. A second aspect which we now further highlight is the influence of approximating a cell by a polygon. We demonstrate that this approximation is responsible for a strong hexatic order on the cellular scale in the considered MDCK data from Armengol-Collade et al. (2023).

It is not clear to me what we should learn about the two tissue models by using *q*_2_ and *q*_6_ to quantify cell shape. Can you clearly formulate one or more conclusions?

What we can learn from the research is a dependence of *qp* on model parameters in the two tissue models is

– \begin{document}$\overline{q_{2}}$\end{document} increases with higher activity or deformability

– \begin{document}$\overline{q_{6}}$\end{document} decreases with higher activity or deformability.

Furthermore, *q*_2_ and *q*_6_ are independent and describe distinct properties. Using these models as a basis to coarse-grain and derive continuous models on the tissue scale, these results indicate that more general p-atic liquid crystal theories should be used and the simplest nematic liquid crystal theories might not be sufficient.

The experimental data and their analysis does not seem to add anything to the work. Do you report only data from independent measurements, or did you consider all images of a monolayer?

As we now also analyze experimental data from Armengol-Collado et al. (2023) which confirm our findings on independency of *q*_2_ and *q*_6_ and also confirm that the proposed nematic-hexatic transition is only specific to the use of *γp* for characterizing the shape, additional experimental data are indeed no longer needed. We, therefore, skip the detailed analysis of this data and only keep the results in Fig 1 and Fig 2 and the corresponding figures in the appendix as illustrating examples.

L13: ”P-atic liquid crystal theories offer new perspectives on how cells self-organize (...)” This is a difficult entry, because the average reader of eLife might not be familiar with p-atic liquid crystals.

We agree that p-atic liquid crystals might not be familiar to the average reader. For this reason we introduce orientational order in the introduction with examples demonstrating that not only nematic, but also tetratic and hexatic order have been identified in tissue and introduce the different symmetries. Furthermore, we provide examples for p-atic liquid crystals from other fields and various references. In the conclusion, we also cite models for p-atic liquid crystal theories. Even if the average reader is not familiar with these theories, it should become evident that nematic order might not be sufficient to describe tissue as other symmetries are present as well.

L32: ”nematic” needs to be introduced.

Nematic order is already explained as rotational order with 180° degrees. The references cited discuss nematic liquid crystals in the context of morphological changes in tissue. We therefore only added a standard text book as reference for liquid crystal theories and refrain introducing it in more detail in the manuscript.

Figure 1: Why do you show the data for *q*_3_, *q*_4_, and *q*_5_, which you do not really consider in this manuscript? Same for Figure 2. Why not combine the two figures? Furthermore, you show *qp* without having defined them yet.

We consider all *p* = 2,3,4,5,6, but focus on *p* = 2,6 in the main text and *p* = 3,4,5 in the appendix. Figures 1 and 2 essentially only introduce the subject and help to relate p-atic order to cell shapes and introduce the methodology to analyze the data. Our conclusion is that all p can be important and should be considered in continuous descriptions of tissue.

Equation 1: The notation is confusing: the domain of integration (*C* or *∂C*) also appears as the variable you integrate.

The equation is correct. The variable of integration is 1 or *H* and the domain of integration is *C* (cell) or *∂C* (cell contour).

L68: ”a snapshot of the considered monolayer of wild-type MDCK cells”. Did you analyse only one monolayer? Please, provide information about the number of monolayers that were imaged and how many cell shapes were analyzed.

We have analyzed one monolayer and have added the missing information.

L86: ”field-specific prefactors” I do not understand what is meant by these.

Different communities, e.g. physics, mathematics, cosmology, .... use different prefactors in the definition. We have removed this statement.

L89: ”Hadwiger’s characterization theorem”. What is this?

This mathematical result is important to claim robustness and stability, it can be found in the cited reference.

L104: ”the essential property is the continuity”. Essential for what?

Essential ”for our purpose” to characterize the shape of cells by a robust method.

L120: ”the theory also guarantees robust description of p-atic orientation for *p* = 3,4,5,6,...” I do not understand what you mean.

The previous examples only consider *p* = 2. However, the cited theoretical results also hold for p = 3,4,5,6,..

Equations (5) and (6): You define *ψp(C)* twice. Are the definitions equivalent? Why do you need both?

This is not a different definition, equation (6) is a reformulation which is more useful for our purpose. But we indeed define *ϑp* twice. We now use a new symbol to distinguish *ϑp* in Equation 7 and 9.

Figure 4: ”The visualization uses rotationally-symmetric direction fields (known as p-RoSy fields in computer graphics (Vaxman et al., 2016)).” I guess that you have used these fields already in Figure 1, so why introduce them only now?

We have moved this comment to Figure 1.

Figure 6: Using a few discrete values cannot illustrate continuity. Also, the ”jump” in *γp* results from deleting a vertex, so I doubt that this is a fair comparison. Still, I think that it is important to point out to the reader that the value *γp* depends on the number of vertices (here, I allow that two edges connected by a vertex are aligned).

We adjusted the caption to make our point more clear. The last image is a triangle and according to the definition of *γp* is, therefore, described by only three vertices. So, it is indeed a fair comparison. The reviewer is right that the value of *γp* has a strong dependency of the number of used vertices, this is exactly the point that we are trying to make with this figure. Also, adding vertices artificially to make *γp* continuous leads to more problems, as the values for *γp* change if we change the number of vertices. But an equilateral triangle should be recognized as an equilateral triangle, no matter if there is an artificial fourth vertex or not. The triangle in our picture and the triangle that the reviewer mentioned (so our triangle with an artificial fourth vertex) both have the shape of an equilateral triangle, yet for one it is |*γ*_3_| = 1.0 and for the other one it is |*γ*_3_| = 0.935.

While we agree on the reviewers statement about continuity, we did not modify the sentence, as the meaning should be clear.

L160: The definition of the center of mass is incorrect as it is not that of an extended object whose contour is defined by a polygon, but only of the set of vertices. In Figure 6 you write ”the choice of the center of mass highly influences the value of *γp*” - is there really a choice of the center of mass? I thought that it was uniquely defined.

We here only repeat the definition from Armengol-Collado et al. (2023) in order to be able to directly compare our analyses with the results presented therein. We adjusted the caption to be more clear.

L166: What is the weighting you refer to in Equation 9?

We apologize, the reference is to Equation 8. We have modified this.

L312: ”Quantifying orientational order in biological tissues can be realized by Minkowsky tensors”. As mentioned above, you do not really use them, but use Equation (7), which can be defined without reference to Minkowski tensors.

Eq. (7) is part of the irreducible representations of the Minkowsky tensor. Therefore the sentence is correct.

L318: I do not quite understand the link between being able (or not) to compare *qp*’s for different values of p and the interpretability of *q*_2_ and *q*_6_. Also, since you introduce *qp*, how can the question about their comparability be a recurrent challenge? Finally, would you agree that even though a comparison between the absolute values of *q*_2_ and *q*_6_ is inappropriate, one can still meaningfully compare relative changes as a parameter is changed or when comparing cells in different conditions?

We have modified the sentence. Furthermore we agree that one can still meaningfully compare relative changes as a parameter is changed, as we do. However, our claim that *q*_2_ and *q*_6_ are independent, does not allow to conclude any kind of nematic-hexatic phase transition. We have now provided further evidence using the published data of Armengol-Collado et al. (2023), which unequivocally supports this statement. We would also like to remark that the detection of a phase-transition requires a single order parameter, which cannot exist as *q*_2_ and *q*_6_ are independent.

We have further explained this in the main text.

Figure 7: The axes are not labeled.

We added the labels.

L359: ”*q*_2_ and *q*_6_ values cluster tightly”, L362 ”*q*_2_ and *q*_6_ values become highly scattered” Please, quantify.

We kept these formulations but have added statistical measures to these qualitative descriptions, see Supplementary Figures to Fig 7 for the distance correlation and the P-values of the distance correlation. These data support our claim of independence.

L362: ”each *q*_2_ value spans a broad range of *q*_6_ values and vice versa, demonstrating their independence”. Please, use a quantitative test of statistical independence.

We have added statistical information by using the distance correlation and statistical tests, see Supplementary Figures to Fig 7. Similar results are obtained for the Pearson correlation and corresponding tests. However, they are not included as the distance correlation is more general.

L371: Please, define *Q*_2_ and *Q*_6_ in the main text.

We have now added the definition to the Materials and methods section.

L420: A reference seems to be missing.

Thanks for pointing this out. This was a formatting error, we only wanted to cite Balasubramaniam et al. (2021).

L425: ”strong dependence of cell shape on cell density”. But *q*_6_ seems to be rather independent of density, see Figure 11. Also, what do you mean by ”strong”? Can you quantify?

The dependency of the cell shape on the cell density is shown in detail in (Eckert et al., 2023). Furthermore, to describe the cell shape the values for all p are needed. So the change in *q*_2_ already indicates a change in the overall cell shape even as *q*_6_ is barely changing. As we excluded these experimental results now in favor of the experimental data also used in Armengol-Collado et al. (2023), we did not add further evaluations regarding cell density.

L453 ”These divergences [nonmonotonic dependence of *γp* on activity or deformability] highlight the limitations of *γp* in capturing consistent patterns”. I am not sure to follow your argument here.

Besides the quantitative differences seen in comparing Fig. 1 and Fig 2 with the corresponding figures in the appendix, these results show qualitative differences. Using a method which is not robust and not continuous leads to qualitative different results. The nonmonotonic dependence of *γp* is specific to the method but not to the underlying physics.

Appendix 2 - Figure 20: It is not clear how to compare this figure to Figure 3e of Armengol-Collado et al 2023. Please, provide more details.

Appendix 2 - Figure 20 (Appendix 3 - Figure 25 in the revised version) and Figure 3e in Armengol-Collado et al. (2023) cannot be directly compared. Fig 3e shows results of experiments and multiphase field simulations for one parameter stetting and Fig 20 results of the active vertex model for various parameter settings. But both are considered using *γp* and Γ_p_. We have added these computation, see Fig. 13, which indeed reproduces the results from Fig 3e. We refrain from considering corresponding plots to Fig 20 for the multiphase field model, as this first requires computing the vertices and no additional information can be expected.

**Reviewer 2:**
The manuscript lacks statistical information. The following should be addressed: How often have the experiments been performed? How many monolayers have been analyzed? How many time steps have been considered and in what duration? How many cells have been included in the analysis? What are the p-values to determine if *qp*’s (Figure 2, panel a) and *γp*’s (Appendix 2-Figure 17, panel a) are significantly different? Same figures: How many cells and experiments have been considered here? Figure 11: What is the density of cells for each condition? Please provide the corresponding values. How significant are the differences? How many times has the experiment been repeated? Figure 12: Due to cell proliferation, the cell density changes over time. Does this need to be taken into account?

We agree, our information have only been qualitative. We have added the missing information. Especially we added statistical information by using the distance correlation and statistical tests, see Supplementary Figures to Fig. 7. Similar results are obtained for the Pearson correlation and corresponding tests (not included). As we excluded the experimental results previously shown in Figure 11 and Figure 12, in the revised version in favor of the experimental data that is already published in Armengol-Collado et al. (2023), we did not add further statistics regarding this. We added the number of frames and cells in the text.

The image analysis part of the Method section states that time-series were xy-drift corrected, and cells were tracked. However, the manuscript does not contain results of dynamical data, timedependent analyses, or discussions of how *qp* changes over time. The authors mention that the fluidity of the tissue was confirmed by the MSD, neighbor number variance, and the self-intermediate scattering function, but none of the results are shown in the manuscript. I would like to ask the authors to provide the results and related content in the Method section.

We have modified the description and removed all parts related to dynamical data. Due to the heavy overload of images in the manuscript we refrain from providing all the results for the phase diagram to distinguish solid and fluid phase. These measures have been provided previously for the considered modeling approaches and provide here only a side remark. Our results do not depend on an exact localization of a solid-fluid phase boundary.

Additional information is missing in the Image analysis part of the Method section. Could the authors provide the information on the image analysis steps between obtaining the segmented image and inputting the parameters for the Minkowski tensor? This should include how the normal vectors have been determined and whether this has been done for all pixels along the contour.

We added further details in the section Extraction of the contour in Experimental setup in Methods and Materials and also provide the code to compute *qp* for segmented images.

The authors have analyzed low-resolution phase contrast images acquired with a 10x objective to experimentally support their introduced Minkowski tensors. This may have decreased the resolution of the cell boundary detection and its curvature. I strongly suggest imaging the tissue with higher magnification (40x or 63x) and/or fluorescent markers to visualize the cell boundaries in high quality. This would allow the authors to distinguish between circles and circle-like shapes (lines 432-434) and to further investigate differences between MDCK wild-type and MDCK E-cad KO cells.

We agree that higher resolution of the images would be beneficial. However, we are convinced that this will not influence our findings. Instead of performing the experiments with higher magnification or using fluorescent markers, we have considered the experimental data from Armengol-Collado et al. (2023) to support our results.

The authors have coarse-grained the shape function, Γ_p_, and have chosen the active vertex model (Appendix 2-Figure 20) for comparison with the Minkowski tensors, *Qp* (Appendix 2 Figure 13). In both figures, the hexatic-nematic crossover does not occur. Armengol-Collado et al. have previously reported that the Voronoi model failed to achieve the hexatic-nematic crossover and argued that this is due to the artificial enhancement of the polygon’s hexagonality, leading to high hexatic order at the tissue scale. Since the authors have used the Voronoi-tailing method (line 196), I would like to ask the authors to compare the multiphase field models for Γ_p_ and*Qp* instead.

We would like to mention that we do not consider a Voronoi model but an active vertex model. A Voronoi model is only used for initialization. Both models are certainly related but not identical and claims for a Voronoi model do not need to hold for an active vertex model. The suggested comparison for the multi phasefield model is not an easy task as it requires to compute the vertices from the phase field variables. There are gaps between cells and a reliable algorithm to identify the vertices is a task on its own. We, therefore, refrain from doing these calculations. Instead, we have used the experimental data from Armengol-Collado et al. (2023) for which the polygonal information are provided, see Figure 11. Especially for *p* = 6, strong differences can be seen by comparing the PDF obtained by the full shape and the polygonal shape. Indeed, the strong hexatic order at the cellular scale is only a consequence of the approximation by polygons. With this result analysing the multi phasefield data by *γp* does not add any new information as this first requires an approximation by polygons.

The authors show the *qp* distributions for the experimental systems (Figure 2, Figure 11). For completeness, I would like to ask the authors to also coarse-grain *qp* and *γp* of the experimental data as shown for the computational models in Appendix 2 - Figure 13 and Appendix 2 - Figure 14. It would be interesting to see if the hexatic-nematic crossover appears. I would recommend that the authors avoid using the Voronoi tailing of the experimental system, as this may fail to obtain the crossover as explained in (5) above. Instead, I suggest using the real vertex positions for *γp*, which can be obtained from the segmented images.

It remains open what is meant by ”the real vertex positions for *γp*, which can be obtained from the segmented images”. Segmenting the images leads to smooth contours, partly even with gaps between cells. As the magnitude of *γp* depends on the number of points used in the calculation it is not meaningful to use all points of the contour for calculating *γp*, as this would lead to artificially low values for |*γp*|. Identifying the vertex positions for an approximating polygon is an issue of its own and the consequence of this approximation is already mentioned above. For a comparison we therefore added the experimental data from Armengol-Collado et al (2023) and used the provided vertex positions to compute *qp* and *γp* as well as the raw data and performed the segmentation and used these data to compute *qp*. See Figure 11. These results confirm our findings and show that the proposed nematic-hexatic phase transition is specific to *γp* to characterize shape.

In order to show that shape descriptors like the shape function, *γp*, introduced by Armengol-Collado et al., ’fail to capture the nuance of irregular shapes’ (line 445), the authors have compared *γp* with the Minkowski tensors, *qp*, using the same dataset (Figure 1 with Appendix 2 - Figure 16, Figure 2 with Appendix 2 - Figure 17, and Figure 4 with Appendix 2 - Figure 15 Appendix 2). I agree that *γp* and *qp* are different, not showing identical values. However, I see no evidence in these figures that *qp* describes the symmetry of a cell better than *γp*, since the values are similar and vary quite similarly between different p-atic orders. What is the quantitative difference that shows the failure of the shape function to capture the nuance of irregular shapes?

The statement already follows from the mathematical properties of robustness and stability, which is illustrated in Fig. 6. The mentioned comparisons for simulation and experimental data only demonstrate that the lack of robustness and stability of *γp* also leads to different results if applied to averages of cell measures. The differences are twofold, first the approximation of cells by polygons leads to different results, and second even for polygons different results follow, as only one approach is continuous and the other not. This has strong consequences for the proposed nematic-hexatic phase transition if coarse-grained. Our added results for the experimental data from Armengo-Collado et al. (2023) show that this behavior is not a physical feature but only specific to the use of *γp*.

The authors claim that the Minkowski tensors provide a ’reliable framework’ and that this framework ’opens new pathways for understanding the role of orientational symmetries in tissue mechanics and development’ (line 78-79). However, the p-atic orders in the experimental systems peak at very low orders of *qp* < 0.3, which may not allow conclusions about (non-)dominant orientational symmetry(ies) of cells. Can this framework be applied to experimental systems? Since the Minkowski tensors display the independence of the hexatic and nematic symmetry, the variations of cell shapes in experimental systems are too strong to provide any additional results (line 437), as stated by the authors, and no crossover was found, while the crossover was reported by Armengol-Collado et al., what new pathways can be opened to study tissues?

We have added a comparison with experimental data from Armengol-Collado et al. (2023) and demonstrate that the proposed nematic-hexatic transition is only specific to the use of *γp* for characterizing the shape. So our results first of all essentially close the ”pathway for understanding the role of orientational symmetries in tissue mechanics and development”, which was proposed on this nematic-hexatic transition. On the other side, even if *qp* peaks at relatively low values, the results demonstrate independence of the measures for different p’s, for two different modeling approaches and two different sets of experimental data. This motivates to consider p-atic order for different p simultaneously. Such theories of ”multi”-p-atic liquid crystals, as proposed in the conclusions, are the mentioned new pathways.

In principle, the introduced Minkowski tensors integrate the orientation of the normal vectors (Equation 6) and consider the perimeter of the contour (Equation 1). Do the tensors distinguish between convex and concave curvature since both are present in tissues? Does a square with 4 concave and a square with 4 convex edges (same curvature) have the same *qp* values?

For the specific situation of a square with 4 concave or 4 convex edges even p would lead to the same orientation and the same value for *qp*, as even p have a 180 degree symmetry. Odd p would result in the same value for *qp* but in a different orientation *ϑp*. In more general cases, e.g. shapes with concave and convex edges, no general statements can be made. In general the theoretical results on stability of *qp* only hold for convex shapes. However, as discussed in Methods and materials the known counterexamples for concave shapes are not relevant for cell shapes.

In lines 169-172 and Figure 6, the authors report a jump in *γp*. Why has the fourth vertex in the last image been removed? The vertices are essential for the calculation of *γp*. If the fourth vertex is not removed, the following values result: *γ*_3_ = 0.935 and *γ*_4_ = 0.474, which leads to changes of the same order of magnitude as those of *qp*. I think it is therefore not the choice of the center of mass that ’heavily influences the value of *γp*’, but the removal of the fourth vertex.

We adjusted the caption to make our point more clear. The last image is a triangle and according to the definition of *γp* is therefore described by only three vertices. The reviewer is right that the value of *γp* has a strong dependency of the number of used vertices, this is exactly the point that we are trying to make with this figure. An equilateral triangle should be recognized as an equilateral triangle, no matter if there is an artificial fourth vertex or not. The triangle in our picture and the triangle that the reviewer described (so our triangle with an artificial fourth vertex) both have the shape of an equilateral triangle, yet for one |*γ*_3_| = 1.0 and for the other one it is |*γ*_3_| = 0.935. This can be seen even more clearly if even more artificial vertices on the outline of the equilateral triangle are added, which will decrease |*γ*_3_| even more. Furthermore, we think there was a misunderstanding regarding our statement about the center of mass. The general problem of *γp* - so the dependence of the values on the number of vertices - is independent of the calculation of the center of mass. The exact values of *γp* on the other hand depend on the choice of this. We follow Armengol-Collado et al. (2023) and use the mean of all vertex coordinates as center of mass. If the reviewer would use the center of mass of the equilateral triangle and do the same calculations the resulting values for *γp* would be different. This is what we meant with ’heavily influences the value of *γp*’.

In Appendix 2 - Figure 18, the authors show that the shape function, *γ*_6_, exhibits a non-monotonic trend as a function of activity and deformability. I have no objection to this statement. However, I would like to ask the authors to check the values for *γ*_6_. In the bottom-left corner, for example, *γ*_6_ = 0.55. This value seems very low to me. In Appendix 2-Figure 20, |*Q*_6_| for *R/Rcell* = 2 is already in this range, while |*Q*_6_| for *R/Rcell* = 1 (not shown), corresponding to *γ*_6_, must be even higher. Also, the parameters *p*_6_ = 3.5 and *v*_0_ = 0.1 should result in a nearly hexagonal lattice, which should be captured with high *γ*_6_ values. I would expect *γ*_6_ to be in the same range as *q*_6_.

Many thanks for pointing this out. There are two different points addressed in this question: The first is if |Γ_p_| is too high. We checked the values, |Γ_p_| = 0.5075 for *R/Rcell* = 2, so it is lower than \begin{document}$\left|\gamma_{6}\right|$\end{document} = 0.58. The second question is why *γp* and *qp* are not in the same value range. You are right that for a perfectly hexagonal lattice both should give the same value, namely \begin{document}$\overline{q_{6}}$\end{document} = *p*\begin{document}$\left|\gamma_{6}\right|$\end{document} = 1.0. However, even at _6_ = 3.5 and *v*_0_ = 0.1 this is not a perfectly hexagonal lattice anymore and how fast the values of *q*_6_ and |*γ*_6_| drop if we move away from a perfect hexagon scales differently. As *qp* is stable and only changes slightly for slight changes in the shape it makes sense, that *qp* is still close to 1.0 . We included an image, see below, of one time step in said parameter to showcase that cells do not form a perfect hexagonal lattice anymore.

**Reviewer 3:**
Could the authors show why and how this method could bring new information which were missing so far in the understanding of morphogenesis in vitro and in vivo with the current quantification?

The introduction provides examples of how orientational order and its topological defects can be linked to morphological changes in tissues. The orientational order emerges from the shape of the cells. Most commonly nematic order has been considered, but more recently also hexatic order and even a nematic-hexactic crossover on larger scales. This suggests a mechanical mechanism for morphogenesis, like a phase transition from hexatic to nematic, which would have consequences on the evolution of shape. We demonstrate that the measures *q*_2_ and *q*_6_ are independent. Furthermore the proposed nematic-hexatic transition is only specific to the use of *γp* for characterizing the shape and coarse-graining of the associated order. These measures are not robust and therefore should not be used. Results for the robust measures *qp* suggest to consider all *p* for a coarse-grained theory to model morphological changes in tissues.

Could authors show quantitative comparisons between available methods with the same sets of data and highlight pros and cons?

**Author response image 1. sa3fig1:** Screenshot from *p*_6_ = 3. 5 and *v*_0_ = 0.1

In addition to what was already done for the simulation data we have added data from Armengol-Collado et al. (2023) and compared the results for *qp* and *Qp* and *γp* and Γ_p_. The theoretical results and the illustrating example in Fig. 6 already show that there are no pros for *γp*. Other methods belong to the class of bond-order methods and measure neighbor relations instead of shape. We already comment that these methods are inappropriate to classify shape, see Methods and materials, last sentence and Mickel et al. (2013) for a detailed discussion why these methods are not robust.

Instead of using phase contrast images, which exhibit curved cell-cell contours, could authors use data with E-cadherin staining instead - as used in many epithelial studies in vitro and in vivo? Could they show both images for wild type and for the E-cadherin KO cell lines with fluorescent readout?

We are convinced that our results do not depend on the way to visualize the cell contours. Furthermore the images do not provide additional information. To further strengthen the experimental part of the manuscript, we instead analyzed data from Armengol-Collado et al. (2023).

They confirm our findings.

The authors acknowledge differences in density between cell lines p. 13 so this calls for new experiments with solid readouts and analysis using comparable experimental conditions.

Additionally, we analyzed data from Armengol-Collado et al. (2023) which confirm our findings. Our results are now supported by two different modeling approaches and two different experimental settings. Because of redundancy we removed the original experimental data from the revised manuscript.